# Determinants of HIV-1 reservoir size and long-term dynamics during suppressive ART

Nadine Bachmann [1,2], Chantal von Siebenthal[1,2], Valentina Vongrad [1,2], Teja Turk [1,2], Kathrin Neumann [1,2], Niko Beerenwinkel [3,4], Jasmina Bogojeska[5], Jaques Fellay [6,7], Volker Roth [8], Yik Lim Kok[1,2], Christian W. Thorball [6], Alessandro Borghesi [6,9], Sonali Parbhoo [8], Mario Wieser [8], Jürg Böni [2], Matthieu Perreau[10], Thomas Klimkait [11], Sabine Yerly [12], Manuel Battegay[13], Andri Rauch [14], Matthias Hoffmann [15], Enos Bernasconi [16], Matthias Cavassini[17], Roger D. Kouyos [1,2,31], Huldrych F. Günthard [1,2,31] & Karin J. Metzner [1,2,31] & the Swiss HIV Cohort Study

The HIV-1 reservoir is the major hurdle to a cure. We here evaluate viral and host characteristics associated with reservoir size and long-term dynamics in 1,057 individuals on suppressive antiretroviral therapy for a median of 5.4 years. At the population level, the reservoir decreases with diminishing differences over time, but increases in 26.6% of individuals. Viral blips and low-level viremia are significantly associated with slower reservoir decay. Initiation of ART within the first year of infection, pretreatment viral load, and ethnicity affect reservoir size, but less so long-term dynamics. Viral blips and low-level viremia are thus relevant for reservoir and cure studies.

[1] Department of Infectious Diseases and Hospital Epidemiology, University Hospital Zurich, 8091 Zurich, Switzerland. [2] Institute of Medical Virology, University of Zurich, 8057 Zurich, Switzerland. [3] Department of Biosystems Science and Engineering, ETH Zurich, 4058 Basel, Switzerland. [4] SIB Swiss Institute of Bioinformatics, 4057 Basel, Switzerland. [5] IBM Research—Zurich, 8803 Rüschlikon, Switzerland. [6] School of Life Sciences, EPFL, 1015 Lausanne, Switzerland. [7] Precision Medicine Unit, Lausanne University Hospital, 1011 Lausanne, Switzerland. [8] Department of Mathematics and Computer Science, University of Basel, 4001 Basel, Switzerland. [9] Neonatal Intensive Care Unit, Fondazione IRCCS Policlinico San Matteo, Pavia 27100, Italy. [10] Division of Immunology and Allergy, Centre Hospitalier Universitaire Vaudois, University of Lausanne, 1015 Lausanne, Switzerland. [11] Division Infection Diagnostics, Department Biomedicine—Petersplatz, University of Basel, 4001 Basel, Switzerland. [12] Division of Infectious Diseases and Laboratory of Virology, University Hospital Geneva, University of Geneva, 1211 Geneva, Switzerland. [13] Department of Infectious Diseases and Hospital Epidemiology, University Hospital Basel, 4031 Basel, Switzerland. [14] Department of Infectious Diseases, University Hospital Bern, 3010 Bern, Switzerland. [15] Division of Infectious Diseases, Cantonal Hospital of St. Gallen, 9007 St. Gallen, Switzerland. [16] Infectious Diseases Service, Regional Hospital, 6900 Lugano, Switzerland. [17] Division of Infectious Diseases, Centre Hospitalier Universitaire Vaudois, University of Lausanne, 1015 Lausanne, Switzerland. [31] These authors contributed equally: Roger D. Kouyos, Huldrych F. Günthard, Karin J. Metzner. A list of participants and their affiliations appears at the end of the paper. Correspondence and requests for materials should be addressed to H.F.G. (email: huldrych.guenthard@usz.ch)

Combination antiretroviral treatment (ART) represents a unique success of modern medicine and reduces morbidity and mortality in the majority of human immunodeficiency virus type 1 (HIV-1)-infected individuals[1,2]. However, lifelong ART is required, because effective treatment does not clear the HIV-1 reservoir, which is the major hurdle to cure[3]. Thus, the search for therapeutic interventions that can eliminate or functionally control the HIV-1 reservoir has high priority. Understanding how HIV-1 persists is crucial for the development of cure strategies.

The latent HIV-1 reservoir is generally accepted as long-lived cells harboring replication-competent HIV-1 in a latent state[4–8]. More broadly and in accordance with the terminology used herein, the HIV-1 reservoir consists of all HIV-1 infected cells independent of the replication competence of the integrated virus genome[9].

The HIV-1 reservoir is established early during primary HIV-1 infection[10]. While the first rapid decay of the HIV-1 reservoir after initiation of ART has been studied extensively[11–15], the decay of the HIV-1 reservoir in individuals on ART and its association to residual viremia has so far only been examined in a few small studies with 30–101 individuals[16–20]. The latent HIV-1 reservoir half-life was estimated to be 44 months, measured by limiting dilution culture assay, in a cohort of 62 individuals who had undetectable viremia for up to 7 years (<50 HIV-1 RNA copies/ml plasma). In the same cohort, the decay of the latent HIV-1 reservoir was slowed down by viral blips (but the effect of blips was not statistically significant)[17]. A cohort of 101 individuals, on ART and virally suppressed for at least 4 years, revealed that total HIV-1 DNA declined slowly with a half-life of 13 years[18]. Another study showed that 31% of individuals did not show a negative slope in the years 4–7 after initiation of ART[19].

Despite suppression of viremia, there is a high interindividual variability in the number of latently HIV-1 infected cells[5,6,10,21]. They can intermittently be activated by antigen recognition or as bystanders in a local inflammatory process[3]. However, the association of low-level viremia and activation of latently HIV-1 infected CD4+ cells has been questioned[22]. "Shock and kill" strategies to deplete the latent HIV-1 reservoir have been explored as a curative approach with the aim of targeting pathways involved in maintaining HIV-1 latency with pharmacological compounds. The underlying assumption is that under suppressive ART, activating the latent HIV-1 reservoir will ultimately result in the destruction of the reactivated cells either by the immune system, additional compounds, or by the cytotoxic effects of HIV-1[23]. However, this approach contradicts the intuition that residual viremia may decelerate the depletion of the latent HIV-1 reservoir by infecting new target cells.

We measure total HIV-1 DNA in longitudinal peripheral blood mononuclear cells (PBMC) samples; a marker for the HIV-1 reservoir found to be sensitive, clinically relevant, and feasible in larger populations[9,24]. Levels of total HIV-1 DNA in PBMC were shown to strongly correlate with the levels of inducible virions[25,26] and viral rebound after treatment interruptions[27,28]. Furthermore, during the first 6 months of ART a disproportionate amount of nonintegrated HIV-1 DNA genomes is lost, suggesting that levels of total HIV-1 DNA after prolonged virus suppression largely represent integrated HIV-1 genomes[29].

We aim to identify the determinants associated with HIV-1 reservoir size and long-term dynamics in the nationwide Swiss HIV Cohort Study (SHCS) by including 1057 individuals on suppressive ART for a median of 5.4 years. We find that, at the population level, the reservoir decreases with diminishing differences over time, but increases in 26.6% of individuals. Viral blips are at the same time significantly associated with a high HIV-1 reservoir and a slow reservoir decay, while low-level viremia is associated with a slow reservoir decay. Thus, viral blips and low-level viremia are relevant parameters to monitor in future reservoir and cure studies.

## Results

**Population wide survey of HIV-1 reservoir during suppressive ART.** To investigate viral and host characteristics that steer HIV-1 reservoir size and long-term dynamics in HIV-1 infected individuals who received suppressive combination antiretroviral therapy (ART) for a median duration of 5.4 years, we analyzed longitudinal total HIV-1 DNA levels of 1057 well-characterized individuals enrolled in the SHCS (Fig. 1a). We focused on the later phases starting after ~1.5 years of ART using systematically collected, stored PBMC.

From the 18,688 individuals enrolled in the SHCS as of December 2014, 1382 individuals fulfilled all inclusion criteria (Fig. 1a). We received ≥3 PBMC samples from 1166 individuals and successfully quantified total HIV-1 DNA in at least the first three time points from 1057 individuals (Fig. 1a). Since technical problems, for instance, failed DNA isolation, were the reason for all unsuccessful quantifications of total HIV-1 DNA, failures to measure total HIV-1 DNA are very unlikely to have introduced a bias to our study population of 1057 individuals. These 1057 individuals represent a real-world setting and are diverse in both their demographics and HIV-1 infection characteristics (Table 1). Notably, we included a substantial number of individuals infected with HIV-1 non-B subtypes ($n = 253$), women ($n = 254$), and individuals of nonwhite ethnicity ($n = 217$, for a breakdown of regions of origin see Supplementary Fig. 1). At initiation of ART the median CD4+ cell count was 203/µl blood (IQR: 94–287) and the median log10 HIV-1 RNA copy number was 4.9/ml plasma (IQR: 4.4–5.4).

To characterize the HIV-1 reservoir size and long-term dynamics beyond the first year of suppressive ART, we measured total HIV-1 DNA in PBMC at 3–4 time points per individual. The median time after initiation of ART was 1.5 years (IQR: 1.3–1.7) at the first sample, 3.5 years (IQR: 3.3–3.7) at the second sample and 5.4 years (IQR: 5.2–5.7) at the third sample. An additional fourth sample was provided by 412 individuals with median 10.0 years (IQR: 8.9–11.5) after initiation of ART.

The median HIV-1 reservoir size 1.5 years after initiation of ART was 2.75 (IQR: 2.40–3.02) log10 total HIV-1 DNA copies/1 million genomic equivalents (copies/mge) (Fig. 2a). On the population level, log10 total HIV-1 DNA levels significantly decreased with diminishing differences over time to 2.59 (IQR: 2.30–2.85; $n = 1057$), 2.53 (IQR: 2.23–2.77; $n = 1057$), and 2.52 (IQR: 2.22–2.7; $n = 412$) median log10 total HIV-1 DNA copies/ mge at 3.5, 5.4, and 10.0 years after initiation of ART, respectively (Fig. 2a, c). A subgroup analysis of the 412 individuals with four available total HIV-1 DNA quantifications yielded qualitatively equivalent results excluding the possibility of a cohort effect (Supplementary Fig. 2). The median decay slope during the years 1.5–5.4 after initiation of ART in the 1057 individuals, calculated using a linear regression, was −0.054 (IQR: −0.109 to 0.002) log10 total HIV-1 DNA copies/mge per year ranging from −0.565 to 0.362 (Fig. 2b), which corresponds to a median half-life of 5.6 years on the linear scale (total HIV-1 DNA copies/mge) assuming first order decay kinetics. However, the reservoir did not reach 50% of its size at the first measurement within the half-life period, since the HIV-1 reservoir decay slowed down over time. Of note, in 281 individuals (26.6%) the total HIV-1 DNA level increased (Fig. 2b). Further, we found that in 66.7% (88/132) of cases a positive increase between the first and second total HIV-1 DNA measurement was followed by an increase between the second and third total HIV-1 DNA measurement. Applying

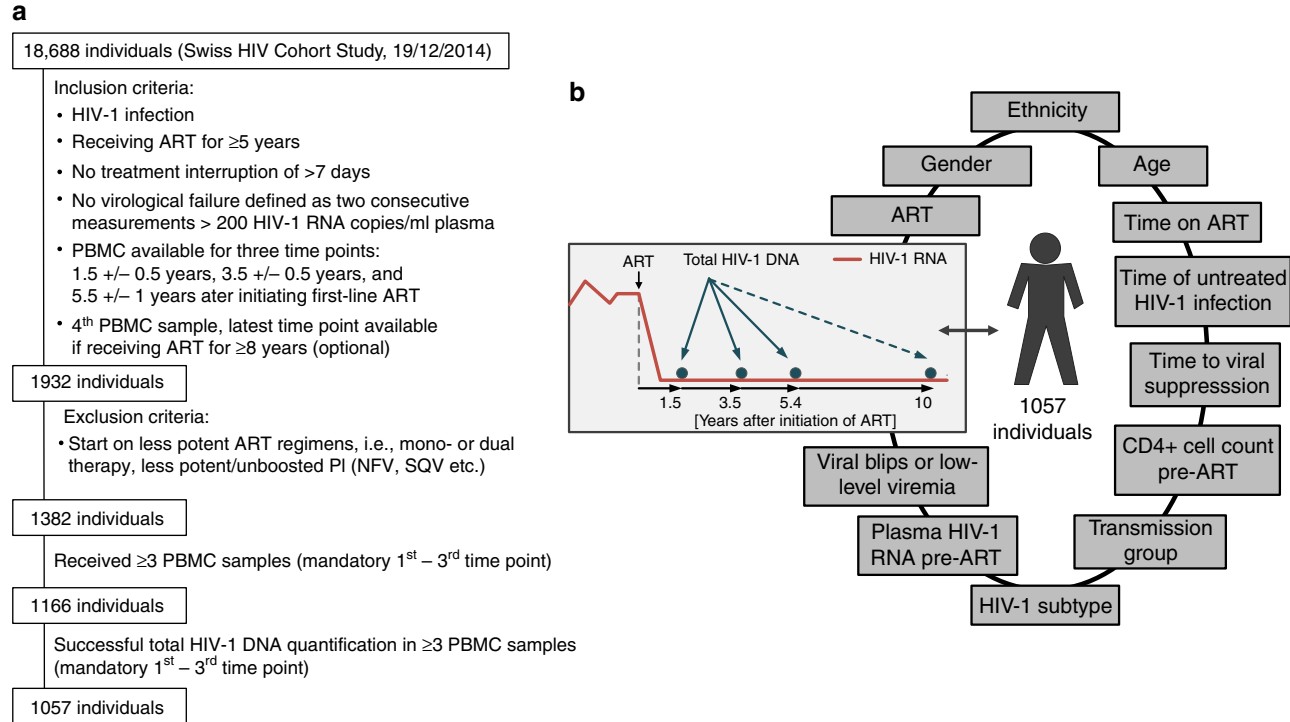

**Fig. 1** Study design. **a** Individuals' selection flow-chart. **b** Overview of the viral and host characteristics considered for an association with HIV-1 reservoir size and long-term dynamics during suppressive ART. ART antiretroviral therapy, PBMC peripheral blood mononuclear cells; time of untreated infection was calculated using estimated dates of infection; time to viral suppression was the time taken for viral load to drop below 50 HIV-1 RNA copies/ml plasma; PI protease inhibitor

this analysis to the 412 individuals providing an additional sample on median 10.0 years after initiation of ART, we calculated a median decay slope of −0.026 (IQR: −0.052 to −0.001) log10 total HIV-1 DNA copies/mge per year ranging from −0.258 to 0.162, which corresponds to a median half-life of 11.8 years on the linear scale (total HIV-1 DNA copies/mge) assuming first-order decay kinetics. Again, in 100 individuals (24.3%) with four measurements available the total HIV-1 DNA level increased (Supplementary Fig. 2c). In summary, there was overall a significant decrease in HIV-1 reservoir size 1.5–5.4 years after initiation of ART. This decrease approached a plateau thereafter.

To study the association of HIV-1 reservoir size and long-term dynamics with various viral and host factors we fitted generalized additive regression models[30] assuming first-order decay dynamics and a Gaussian error distribution. We focused on two distinct characteristics: (i) the total HIV-1 DNA level 1.5 years after initiation of ART, i.e., the HIV-1 reservoir size (Fig. 3a) and (ii) the decay of total HIV-1 DNA from 1.5 to 5.4 years after initiation of ART, i.e., the HIV-1 reservoir long-term dynamics (Fig. 3b).

**Determinants of HIV-1 reservoir size.** To assess the association of various host and viral factors on HIV-1 reservoir size, we focused on the 1.5 years after ART initiation. Viral blips 0.5–1.5 years after ART initiation occurred in 130 (12.5%) individuals (Table 1) and were associated with a larger HIV-1 reservoir 1.5 years after ART initiation ($p = 0.0068$, Fig. 3a). Numerically combining viral blips and low-level viremia as cumulative plasma HIV-1 RNA divided by the number of viral load measurements enhanced the significance of the association with a larger HIV-1 reservoir size 1.5 years after initiation of ART ($p = 0.0011$, Supplementary Fig. 3). We next tested the impact of the height of viral blips on the HIV-1 reservoir size. Viral blips between 50 and

199 HIV-1 RNA copies/ml plasma occurred in 126 individuals within 0.5–1.5 years after initiation of ART and were significantly associated with a larger HIV-1 reservoir size ($p = 0.0037$, Supplementary Fig. 4).

In addition to viral blips, high viral load pre-ART ($p = 0.0087$) and slower time to viral suppression ($p = 0.0062$) were found to be independently associated with a larger HIV-1 reservoir size (Fig. 3a). The opposite effect of CD4+ cell counts pre-ART was observed in the univariable model (Fig. 3a). Even though the median time on ART was 1.5 years and variations were only of small magnitude (IQR: 1.3–1.7), longer time on ART was a significant predictor of a small HIV-1 reservoir ($p = 0.023$, Fig. 3a).

We also found that initiation of ART within the first year of HIV-1 infection ($n = 173$ (16.4%), Table 1) was associated with smaller HIV-1 reservoir size 1.5 years after initiation of ART ($p = 0.0001$, Fig. 3a). For a subset of 16 individuals with known Fiebig stages[31] at the day of initiation of ART, the median HIV-1 reservoir was lower for individuals in Fiebig stages II and IV-VI as compared to the population level median (Supplementary Fig. 5). In particular, the HIV-1 reservoir was strikingly low for two individuals in Fiebig stage II. Initiation of ART beyond the first year after HIV-1 infection was associated with larger HIV-1 reservoir size, independent of the time period of untreated HIV-1 infection analyzed (Supplementary Fig. 6).

Next, we investigated the influence of HIV-1 subtype and host ethnicity on the HIV-1 reservoir size 1.5 years after initiation of ART. HIV-1 subtype was determined by population sequencing in the context of routine genotypic HIV-1 drug resistance testing or retrospective sequencing of stored plasma samples. 253/880 (28.8%) individuals with available HIV-1 subtype information were infected with HIV-1 non-B subtypes (Table 1). HIV-1 subtype was available for 166/217 (76.5%) individuals with nonwhite ethnicity. Of those, 120 (72.3%) individuals were infected with HIV-1 non-B subtypes. Nonwhite ethnicity was significantly associated with a smaller

**Table 1 Individuals' characteristics**

| | | |
|---|---|---|
| Total number of individuals | | 1057 |
| Number individuals with $n$ samples (%) | 3 | 645 (61.0) |
| | 4 | 412 (39.0) |
| Age at first HIV-1 DNA sample in years (median [IQR]) | | 41.0 [35.0, 48.0] |
| Ethnicity (%) | White | 840 (79.5) |
| | Non-white | 217 (20.5) |
| Sex (%) | Male | 803 (76.0) |
| | Female | 254 (24.0) |
| Transmission group by sex (%) | MSM | 524 (49.6) |
| | HET male | 209 (19.8) |
| | HET female | 196 (18.5) |
| | PWID male | 54 (5.1) |
| | PWID female | 28 (2.6) |
| | Other male | 22 (2.1) |
| | Other female | 24 (2.3) |
| Time of untreated HIV-1 infection in years (%) | <1 | 173 (16.4) |
| | 1–3 | 285 (27.0) |
| | 3–5 | 123 (11.6) |
| | 5–7 | 280 (26.5) |
| | >7 | 196 (18.5) |
| Time on ART at first HIV-1 DNA sample in years (median [IQR]) | | 1.49 [1.28, 1.70] |
| Time from ART initiation to below <50 HIV-1 RNA copies/ml in years (median [IQR]) | | 0.33 [0.22, 0.49] |
| CD4+ cell count pre-ART/μl blood (median [IQR]) | | 203.0 [93.5, 287.0] |
| Log 10 HIV-1 plasma RNA pre-ART/ml plasma (median [IQR]) | | 4.91 [4.38, 5.39] |
| HIV-1 RNA (180 days after ART initiation - 1st HIV-1 DNA sample) (%) | <50 copies/ml | 822 (78.8) |
| | Viral blips | 130 (12.5) |
| | Low-level viremia | 91 (8.7) |
| HIV-1 RNA (first to third HIV-1 DNA sample) (%) | <50 copies/ml | 712 (67.4) |
| | Viral blips | 260 (24.6) |
| | Low-level viremia | 85 (8.0) |
| First ART regimen (%) | NNRTI based | 555 (52.5) |
| | Boosted PI based | 502 (47.5) |
| HIV-1 subtype available (%) | | 880 (83.3) |
| HIV-1 subtype (%) | B | 627 (71.2) |
| | 01_AE | 65 (7.4) |
| | 02_AG | 41 (4.7) |
| | A | 39 (4.4) |
| | C | 36 (4.1) |
| | Recombinant | 27 (3.1) |
| | D | 12 (1.4) |
| | F | 12 (1.4) |
| | G | 12 (1.4) |
| | 06_CPX | 2 (0.2) |
| | 11_CPX | 2 (0.2) |
| | 18_CPX | 1 (0.1) |
| | 12_BF | 1 (0.1) |
| | 19_CPX | 1 (0.1) |
| | 20_BG | 1 (0.1) |
| | H | 1 (0.1) |

The time of untreated HIV-1 infection was calculated using the estimated HIV-1 infection dates. Pre-ART log10 HIV-1 RNA copies/ml plasma and pre-ART CD4+ cell count/μl blood refer to the last laboratory values available before initiation of ART. Transmission group stratified by sex indicates the self-reported route of infection (men who have sex with men (MSM), heterosexual (HET), people who inject drugs (PWID), and other (including unknown, and transfusions)). The HIV-1 subtypes were determined using population sequencing. *ART* antiretroviral therapy, *NNRTI* nonnucleoside reverse-transcriptase inhibitors, *boosted PI* boosted protease inhibitor

HIV-1 reservoir size ($p = 0.02$), while HIV-1 non-B subtype only showed a significant association with a small HIV-1 reservoir in the univariable model (Fig. 3a). Using Wilcoxon rank sum test we observed that individuals infected with HIV-1 subtype C showed lower total HIV-1 DNA levels than individuals infected with HIV-1 subtype B ($p = 0.002$, Supplementary Figs. 7a and 8a). Furthermore, the effect of HIV-1 non-B subtypes was still present when we restricted our analysis to individuals with white ethnicity ($p = 0.017$, Supplementary Fig. 7b) and even in a multivariable analysis a trend of a lower HIV-1 reservoir size in individuals infected with HIV-1 non-B subtypes persisted ($p = 0.059$, Supplementary Fig. 9a). We excluded PCR artifacts for this observation by analyzing primer and probe binding sites of the droplet digital PCR assay for 512/1057 individuals for which HIV-1 full-length sequences representing all HIV-1 subtypes were available and no droplet digital PCR-impairing mismatches were detected.

In terms of transmission groups, people who inject drugs (PWID) are associated with smaller HIV-1 reservoir size 1.5 years after initiation of ART ($p = 0.0057$ for female, $p = 0.0012$ for male, Fig. 3a). PWID are associated with a higher incidence of hepatitis C virus (HCV) infection[32], and subsequent interferon treatment might have an impact on HIV-1 reservoir size[33–36]. Therefore, we corrected for interferon treatment for HCV infection (given to 11 individuals prior to the first sampling time point) prior to the measurement of the HIV-1 reservoir size. The smaller HIV-1 reservoir size in the PWID group remained significant after correction for interferon treatment ($p = 0.0051$ for female, $p = 0.0008$ for male, Supplementary Fig. 10). Of note, a substantial fraction of the PWID group was HIV-1 infected years before potent ART became available (Supplementary Fig. 11). Thus, the association PWID with smaller HIV-1 reservoir size 1.5 years after initiation of ART might reflect a survival bias, i.e., an over-representation of long-term nonprogressors among the PWID group, known to have smaller HIV-1 reservoir sizes[37].

**Determinants of HIV-1 reservoir long-term dynamics**. The association of viral and host factors with HIV-1 reservoir long-term dynamics was explored within 1.5 and 5.4 years after initiation of ART. Linear regression slopes of the first three longitudinal total HIV-1 DNA measurements were calculated and corrected for the total HIV-1 DNA level 1.5 years after initiation of ART using a spline to allow for more flexibility (Fig. 3b). The most prominent predictors of a slow decay of total HIV-1 DNA were viral blips and low-level viremia observed within 1.5 and 5.4 years after initiation of ART ($p < 0.0001$ and $p = 0.0074$, Fig. 3b), which occurred in 260 (24.6%) and 85 (8.0%) individuals, respectively (Table 1). Numerically combining viral blips and low-level viremia enhanced the significance of their association with a slower HIV-1 reservoir decay ($p < 0.0001$, Supplementary Fig. 3). Viral blips as low as between 50 and 199 HIV-1 RNA copies/ml plasma, occurring in 236 (22.3%) individuals, were significantly associated with a slower HIV-1 reservoir decay ($p = 0.0022$, Supplementary Fig. 4). Viral blips of 200–499 HIV-1 RNA copies/ml plasma within 1.5–5.4 years after initiation of ART were observed in 48 (4.5%) individuals and were also associated with a slower HIV-1 reservoir decay ($p = 0.0068$, Supplementary Fig. 4). This supports the robustness of viral blips as a predictor of reservoir long-term dynamics.

Considering the established proxies of disease progression, we found that high CD4+ cell counts pre-ART were associated with a faster reservoir decay ($p = 0.0202$, Fig. 3b). Although time to viral suppression was associated with HIV-1 reservoir size, no association was observed with the subsequent HIV-1 reservoir long-term dynamics (Fig. 3).

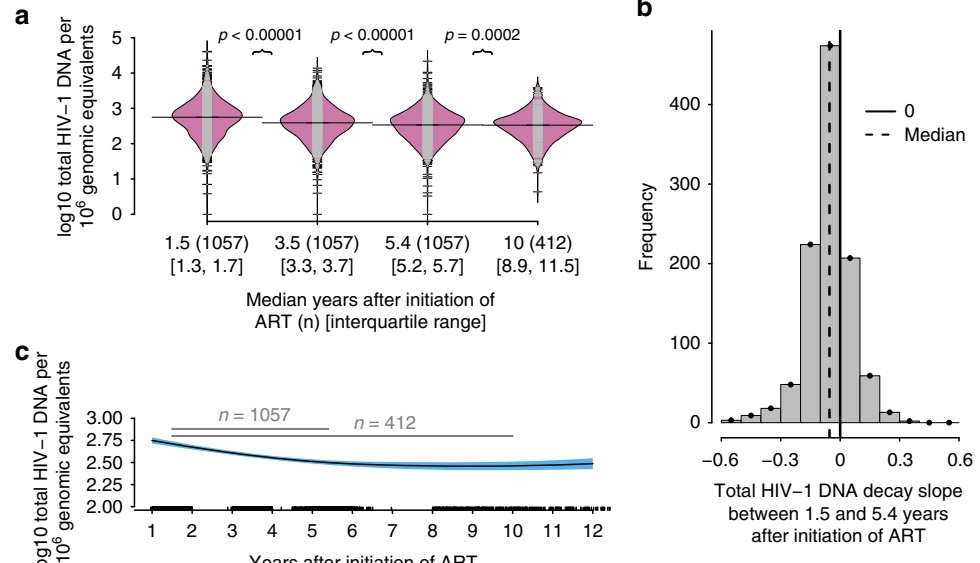

**Fig. 2** The HIV-1 reservoir size and long-term dynamics in 1057 individuals on suppressive ART for on median 5.4 years. **a** Beanplot of total HIV-1 DNA levels in 1057 individuals on long-term suppressive ART at 3–4 different time points (with median 1.5, 3.5, 5.4, and 10.0 years ($n = 412$) after initiation of ART) and the respective sample size. The $p$ values were calculated using paired Wilcoxon tests. The individual observations are shown as small lines (gray or black) in a one-dimensional scatter plot. Overlaid is the estimated density of the distributions (filled in pink) and the median is depicted by a black line. **b** Histogram of linear regression slope over the first three measurements of total HIV-1 DNA levels with median 1.5–5.4 years after initiation of ART. **c** Spline fitted to all log10 total HIV-1 DNA/1 million genomic equivalents showing the 95% confidence intervals in blue and sampling times after initiation of ART in years on the x-axis. Gray lines indicate median observation times for respective number of individuals; ART antiretroviral therapy

Individuals initiating ART within the first year of HIV-1 infection ($n = 173$, 16.3%, Table 1) showed a trend toward a faster decay of the HIV-1 reservoir in the univariable model than individuals starting after the first year of HIV-1 infection ($n = 884$, 87.6%, Table 1). This was not observed in the multivariable analysis (Fig. 3b).

Next, we investigated the influence of HIV-1 subtype and host ethnicity on the HIV-1 reservoir long-term dynamics. HIV-1 non-B subtype was associated with a faster decay of the HIV-1 reservoir in the multivariable analysis ($p = 0.0485$, Fig. 3b). The effect was not only driven by individuals infected with HIV-1 subtype C ($p = 0.1186$, Supplementary Fig. 8). Notably, nonwhite ethnicity showed a trend towards slower decay of the HIV-1 reservoir ($p = 0.075$, Fig. 3b) and was at the same time significantly associated with having a positive slope in a logistic regression model (OR = 1.657, $p = 0.036$, Supplementary Fig. 12).

We further verified our findings of viral and host factors and their association with HIV-1 reservoir long-term dynamics using mixed effect models (Supplementary Fig. 13). Additionally, we showed that our findings were neither driven by the respective ART regimen of the individuals (Supplementary Fig. 14), which was changed frequently (Supplementary Fig. 15), nor by the self-reported adherence (Supplementary Fig. 16).

**Interplay of residual viremia with HIV-1 reservoir size and decay.** Our data reflect a highly complex interplay between residual viremia during suppressive ART and HIV-1 reservoir size and long-term dynamics. On the one hand, viral blips or low-level viremia during suppressive ART were associated with slower decay of the HIV-1 reservoir (Fig. 3b). The predicted log10 total HIV-1 DNA slope is 1.79-fold and 1.93-fold less steep in individuals with viral blips and low-level viremia, respectively, compared to individuals with viral loads constantly below 50 HIV-1 RNA copies/ml plasma (Fig. 4a). Furthermore, we analyzed the predictive value of the degree of residual viremia on HIV-1 long-term dynamics. We observed a strong association between the conditional predicted

log10 total HIV-1 DNA slope and each individual's mean log10 plasma HIV-1 RNA (Fig. 4b). On the other hand, we observed that the HIV-1 reservoir size 1.5 years after ART initiation was strongly predictive for viral blips or low-level viremia during the follow-up period (Fig. 4c). Interestingly, the frequency of viral blips was decreasing with time after initiation of ART and was generally lower in individuals initiating ART within the first year of HIV-1 infection (Supplementary Fig. 17).

These findings highlight the complex interplay between the size and long-term dynamics of the HIV-1 reservoir and residual viremia. The HIV-1 reservoir size is associated with both, subsequent residual viremia and the subsequent HIV-1 reservoir decay, while simultaneously residual viremia inhibits the HIV-1 reservoir decay (Fig. 4c). Given these observations, residual viremia could potentially replenish the HIV-1 reservoir levels while individuals are on long-term ART via two conceivable biological mechanisms as depicted in Fig. 4d.

## Discussion

This study examined the HIV-1 reservoir size and its long-term dynamics over extensive follow-up periods in a well-characterized population-based cohort of more than 1000 individuals. We found a small but continuous decay of the HIV-1 reservoir from years 1.5 to 10 after initiation of suppressive ART. The decay slowed down over time and seems to approach a plateau. Interestingly, 26.6% of our study population exhibited a positive slope of total HIV-1 DNA levels over a median of 5.4 years of suppressive ART. Our study design allowed us to perform a detailed analysis of various viral and host factors. In particular, we assessed the relationships of residual viremia under successful ART with HIV-1 reservoir size and long-term dynamics. Our analysis showed that (i) the presence of viral blips during the first 1.5 years of suppressive ART correlated with HIV-1 reservoir size 1.5 years after initiation of ART and (ii) the presence of viral blips between 1.5 and 5.4 years of suppressive ART or low-level viremia were associated with a slower decay of the HIV-1 reservoir.

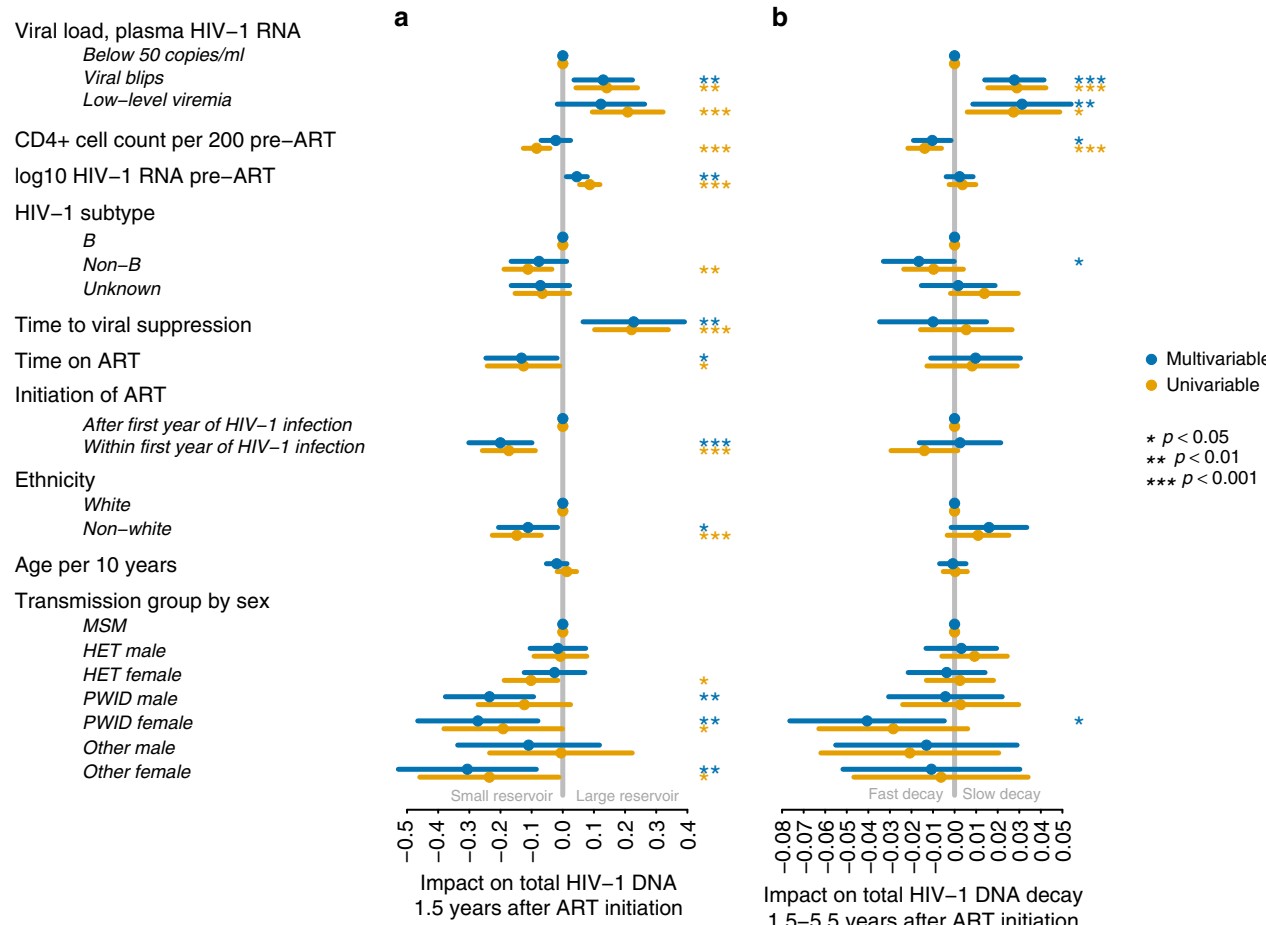

**Fig. 3** Determinants of HIV-1 reservoir size and long-term dynamics. **a** Coefficient plot showing covariables associated with total HIV-1 DNA levels 1.5 years after initiation of ART and 95% confidence intervals. Viral load <50 HIV-1 RNA copies/ml plasma or low-level viremia refer to the time from 180 days after initiation of ART to the first HIV-1 DNA quantification. Reference was defined as viral load, plasma HIV-1 RNA below 50 copies/ml, initiation after first year of HIV-1 infection, transmission group MSM, white ethnicity and HIV-1 subtype B. **b** Coefficient plot showing covariables associated with the decay of total HIV-1 DNA levels and 95% confidence intervals. Corrected for initial HIV-1 DNA levels using a spline. Viral load <50 HIV-1 RNA copies/ml plasma or low-level viremia or viral blips refer to the time between the first and third sample, i.e., 1.5–5.4 years after initiation of ART. Baseline as in panel (**a**). ART antiretroviral therapy, MSM men who have sex with men, HET heterosexual, PWID people who inject drugs, transmission group other includes unknown, and transfusion; time to viral suppression refers to time taken for viral load to drop below 50 HIV-1 RNA copies/ml plasma; CD4+ cell count was measured per 200 cells/μl blood

The interplay of viral blips and total HIV-1 DNA levels is complex, and the origin of viral blips is thought to be multifactorial. (i) It could reflect intermittent virus production of activated latently HIV-1 infected cells[38]. (ii) It might indicate transient episodes of increased viral replication, e.g., by increased availability of activated target cells that could be infected by virus replicating at low levels in sanctuary sites such as lymphatic tissues that have been suggested to be less permissive to antiretroviral drugs[38–40], or by periods of diminished adherence[41,42]. (iii) Viral blips could signify clonal expansion of temporarily virus-producing, but otherwise latently HIV-1 infected cells[43–45]. (iv) Assay and sample preparation variability may potentially account for a fraction of the viral blips observed[42]. In our study, lack of adherence is unlikely to be a major driver of viral blips since virologic failure and treatment interruptions were exclusion criteria and self-reported adherence was high in the study population as 920/1051 (87%) individuals never reported to have missed more than 1 pill in the previous month at all bi-annual SHCS follow-up visits between the first and third time point. Despite being self-reported (and hence subject to potential biases), the adherence measure reported in the SHCS is a validated prediction marker for important clinical outcomes such as

virological failure and mortality[46]. Furthermore, we found that upon adjusting with the proxy of adherence, the effect-sizes for viral blips and low-level viremia changed only marginally (Supplementary Fig. 16). Notably, the HIV-1 reservoir size 1.5 years after initiation of ART was also associated with prior viral blips and predictive for viral blips thereafter, and subsequently associated with a slower decay of the HIV-1 reservoir. This argues for a biological relationship of these factors and rules out a major impact of assay variability. Are viral blips a consequence of the HIV-1 reservoir size? Are they a reason for slowing down the decay of latently HIV-1 infected cells, i.e., are they contributing towards the persistence of HIV-1? Initiation of ART within the first year of HIV-1 infection reduces the HIV-1 reservoir significantly when compared to treatment initiation during chronic HIV-1 infection[13,15,20,47–50]. Consistently, we also observed this effect in our study. The HIV-1 reservoir size and the frequency of viral blips during suppressive ART in the first 1.5 years and thereafter were significantly lower in individuals treated within the first year of HIV-1 infection when compared to individuals initiating ART after the first year of HIV-1 infection. This suggests that the HIV-1 reservoir size can explain the frequency of viral blips. This finding

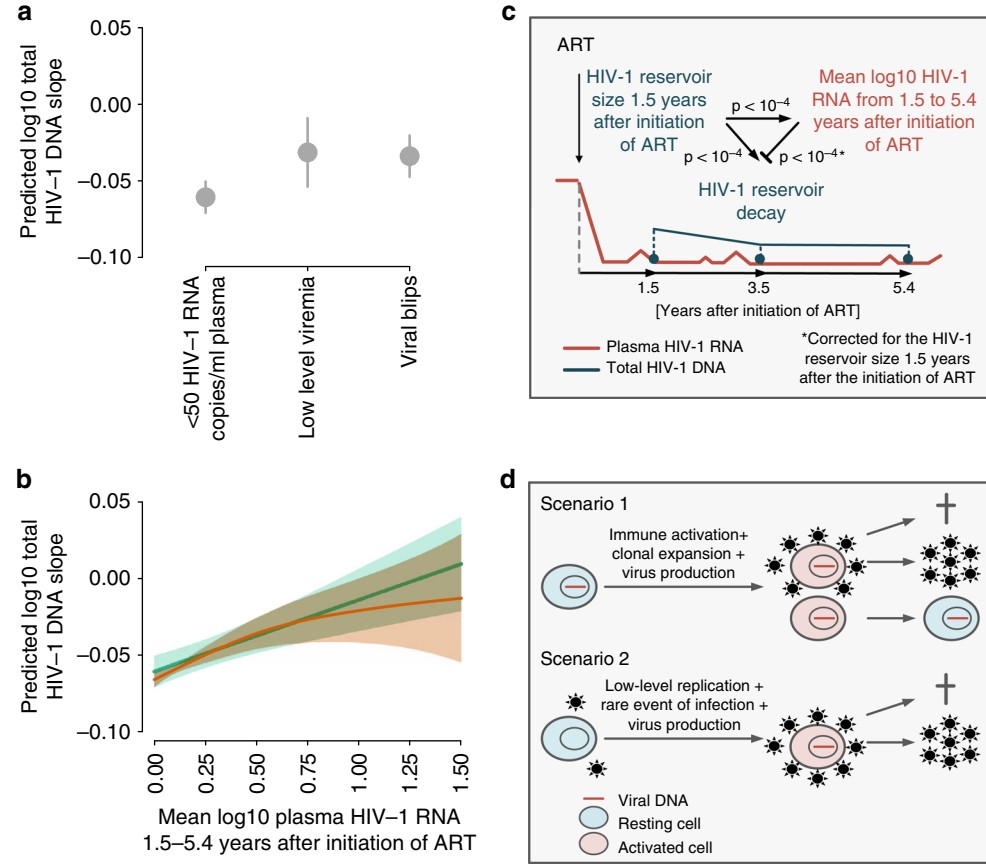

**Fig. 4** The interplay of residual viremia and HIV-1 reservoir size and long-term dynamics. **a** The predicted log10 total HIV-1 DNA slope conditional on the effect of all other covariables and the observed viral load (constantly < 50 HIV-1 RNA copies/ml plasma, occurrence of viral blips, or low-level viremia) within 1.5–5.4 years after initiation of ART. **b** The predicted log10 total HIV-1 DNA slope and 95% confidence interval conditional on the effect of all other previously included covariables and the observed mean log10 plasma HIV-1 RNA (viral blips and low-level viremia) within 1.5–5.4 years after initiation of ART is shown in green. This was calculated using the R-package margins. The result of fitting a spline as the smoothing function for mean log10 plasma HIV-1 RNA (viral blips and low-level viremia) within 1.5–5.4 years after initiation of ART while correcting for the previously included covariables is shown in red. We choose a thin plate regression spline with dimension 20 of the basis. **c** Conceptual figure showing the observed associations between: (i) residual viremia, (ii) the HIV-1 reservoir size 1.5 years after initiation of ART, and (iii) the decay of the HIV-1 reservoir 1.5–5.4 years after initiation of ART. Residual viremia captured both, low-level viremia and viral blips, from 1.5 to 5.4 years after initiation of ART and was thus indicated as the mean log10 HIV-1 RNA. Arrows indicate a positive effect size, i.e., an enhancing effect. The blunted arrow indicates an inhibiting effect. *p* values were derived using linear regression. ART antiretroviral therapy. **d** Two possible scenarios explaining the interplay between residual viremia and HIV-1 reservoir size and long-term dynamics

supports the possibility that viral blips (i) may partially reflect transient episodes of viral replication, which may intermittently refill the HIV-1 reservoir or (ii) mirror the expansion of clones of latently HIV-1 infected cells exhibiting intermittent virus production[51]. The absence of viral evolution in most studies, which investigated HIV-1 genomes from latently infected cells from the peripheral blood, argues against the first hypothesis in well-suppressed individuals[52]. However, whether evolutionary processes may be ongoing in sanctuary sites is less well studied[51]. A possible effect of viral blips on the latent HIV-1 reservoir has been assessed in two studies, which reported decay differences in groups of individuals with and without blips[16,17]. Furthermore, there is evidence that viral blips may be biologically meaningful, since a dose response of viral blip magnitude and virologic failure in fully adherent individuals has been reported[42].

The size of the latent HIV-1 reservoir at 1.5 years after initiation of ART as measured by total HIV-1 DNA in our large cohort was also strongly governed by time of ART initiation. However, there was no significant effect of time of initiation of ART on the total HIV-1 DNA decay. This confirms previous results, which showed that, although earlier ART was associated with a faster total HIV-1 DNA decay during the first 8 months, this effect was lost on the decay rate of HIV-1 DNA at the subsequent phase[15].

An interesting finding of our study was the independent and significant effect of nonwhite ethnicity on the size of total HIV-1 DNA levels 1.5 years after initiation of ART. This result is consistent with Müller et al.[53] who reported lower viral set-points in people from African origin and Gossez et al.[54] who found a higher probability of virological remission following treatment interruption among female Africans compared to female non-Africans. Although not significant, we observed trends of lower HIV-1 reservoir size and faster HIV-1 reservoir decay in individuals infected with HIV-1 non-B subtypes even after controlling for ethnicity. These findings accentuate the need to further explore viral subtype- and ethnicity-dependent pathogenetic mechanisms[55–57]. Notably, total HIV-1 DNA levels did not decrease in 281/1057 (26.6%) individuals during the years 1.5–5.4 after initiation of suppressive ART. This is in agreement with Besson et al.[19], who reported that 9 out of 29 (31%) individuals did not show a negative HIV-1 DNA slope in the years 4–7 after initiation of ART.

Our study is not without limitations. As with any observational study, the reported associations cannot be assumed to reflect causal relationships. They must be interpreted cautiously because of the potential for unmeasured confounding factors. However, the extensive data of the underlying SHCS did allow us to correct for many known and suspected confounding factors. This study examined over 1000 well-characterized HIV-1 infected individuals with 3–4 samples per individual. The number of cells per sample, however, was insufficient for further dissection of the various CD4+ cell subsets[58,59]. We modeled the HIV-1 reservoir decay as linear on the log10 scale based on three time points. Together with potential assay and within-subject sample variability, this indicates some uncertainty around individual slopes. Despite these and other potential limitations (e.g., not differentiating between replication competent- and -defective viruses), our study shows that total HIV-1 DNA measured in PBMC is a robust proxy for the latent HIV-1 reservoir size after the first rapid decay following initiation of ART for several reasons. It correlates independently with time to initiation of ART, with pretreatment viral load, pretreatment CD4 count, and with viral blips. In addition, the decay of the latent HIV-1 reservoir is in line with smaller studies using either viral outgrowth assays or total HIV-1 DNA[17,19]. Furthermore, if slopes were due to within-subject sample variability, a lower fraction of consistent within-subject increase/decrease would have been observed.

A significant strength of our study is that it reflects individuals enrolled in a highly representative cohort study, as 75% of all treated HIV infected individuals in Switzerland are enrolled in the SHCS[60]. Utilizing this unique cohort of >1000 individuals, we demonstrate an interplay between total HIV-1 DNA levels, viral blips and low-level viremia, and the decay of total HIV-1 DNA in well suppressed patients. This suggests that viral blips and low-level viremia are relevant parameters to monitor in future reservoir and cure studies. The very slow and decelerating HIV-1 DNA decay after on median 5.4 years also reiterates the need for targeted new interventions to reduce the HIV-1 reservoir, because long-term ART seems not to substantially affect it.

## Methods

**Study design and study participants.** The SHCS is a nation-wide, prospective observational study founded in 1988 enrolling HIV-infected adults of all transmission groups. Clinical and laboratory data are collected every 3–6 months and plasma and cell samples are stored every 6–12 months. More than 75% of all HIV-1 infected individuals living in Switzerland and receiving ART are enrolled in the SHCS[60]. The current study participants were included when they fulfilled the following inclusion criteria: (1) start on potent ART regimen (i.e., no mono- or dual therapy, no less potent/unboosted PI (NFV, SQV, etc.), (2) no treatment interruption of >7 days, (3) no virologic failure as defined by two consecutive viral load measurements >200 HIV-1 RNA copies/ml plasma, and (4) available cell samples during ART (Fig. 1a). The SHCS has been approved by the ethics committee of the participating institutions (Kantonale Ethikkommission Bern, Ethikkommission des Kantons St. Gallen, Comite departemental d'ethique des specialites medicales et de medicine communautaire et de premier recours, Hôpitaux Cantonale de Genève, Kantonale Ethikkommission Zürich, Repubblica e Cantone Ticino—Comitato Etico Cantonale, Commission cantonale d'étique de la recherche sur l'être humain, Canton de Vaud, Lausanne, Ethikkommission beider Basel) and written informed consent had been obtained from all participants.

**Cells.** Cryopreserved PBMC from 3 to 4 longitudinal time points from each individual during ART were collected from the SHCS biobank: first time point ~1.5 years, second time point ~3.5 years, third time point ~5.5 years, and an optional fourth time point >8 years after the initiation of ART.

As negative controls, PBMC from HIV-1 negative donors or Sup-T1 cells were used. As HIV-1 positive controls, negative controls were spiked with ACH-2 or J-Lat (clone 10.6) cells in a 1000:1 ratio. Batches of negative and positive controls were cryopreserved before genomic DNA extraction. All cell lines were obtained through the NIH AIDS Reagent Program, Division of AIDS, NIAID, NIH. PBMC were derived from buffy coats obtained from healthy blood donors, as anonymously provided by the Blood Donation Service Zurich, Swiss Red Cross, Schlieren, Switzerland. Written consent for the use of buffy coats, which were not

required for medical treatment, but for research purposes was obtained from blood donors by the Blood Donation Center.

**Genomic DNA extraction and fragmentation.** Total genomic DNA was extracted from approximately $2–3 \times 10^6$ cells. DNA extraction was performed manually or by means of the QIAcube using the DNeasy Blood & Tissue Kit (QIAGEN) according to the manufacturer's instruction using a customized protocol as the following: DNA on columns were washed twice with buffer AW2, centrifuged twice after the last washing step, and eluted with $H_2O$ preheated to 70 °C. DNA concentration was measured by the A260/A280 absorptivity ratio using a NanoDrop 1000 spectrophotometer (Thermo Scientific). Totally, 50 μl DNA was sheared to mean size fragments of 1 kB using a M220 Focused-ultrasonicator (Covaris) according to the manufacturer's instructions.

**Quantification of total HIV-1 DNA by droplet digital PCR.** The RainDrop™ digital PCR system (RainDance Technologies Inc.) was used for the quantification of total HIV-1 DNA. Briefly, approximately 3.3 μg DNA in 33 μl water was added to 17 μl PCR master mix, i.e., the final PCR contained 1× PCR buffer (Sigma-Aldrich), 3 μM $MgCl_2$ (Sigma-Aldrich), 0.4 μM PCR-grade dNTP mix (QIAGEN), 1× Droplet stabilizer (RainDance Technologies Inc.), 0.8 μM each of the oligonucleotides (Microsynth) mf51_655 5′-TGCAGCTCTCATTTTCCATAC-3′ (nt 586–606, CCR5 gene, Genbank accession number U54994), mf52 5′-GAGTTTTTAGGATT CCCGAGTA-3′ (nt 694–715, CCR5 gene), HIVfw4924 5′-ACTTTGGAAAGGAC CAGC-3′ (nt position 4928–4945, HIV-1$_{HXB2}$, Genbank accession number K03455), HIVrev5004 5′-CTTTTCTYCTTGGYACTAC-3′ (nt position 5004–5022, HIV-1$_{HXB2}$), 0.2 μM PrimeTime® LNA–ZEN probe (Integrated DNA Technologies) Mf73tq 5′-TET/CCGCTGCTT/ZEN/GTCATGGTCATCTG/3′IABkFQ (nt 668–692, CCR5 gene), 0.13 μM PrimeTime® LNA–ZEN probe aHIVas 5′-6FAM/ TGCCCCTTC/ZEN/ACCTT + TCCA/3′IABkFQ (nt position 4956–4973, HIV-1$_{HXB2}$), 0.06 μM PrimeTime® LNA–ZEN probe aHIV2nd 5′-6-FAM/CA + G + T + A + G TAA TAC AA + G ATA ATA + GTG /3′IABkFQ (nt position 4972–4995, HIV-1$_{HXB2}$), and 0.05 units/μl JumpStart™ Taq DNA Polymerase (Sigma-Aldrich).

Up to eight samples were loaded in parallel on a RainDance Source chip. All three to four samples of each individual were processed and analyzed in the same run. Automated emulsion-based droplet generation was performed in the RainDance Source instrument according to the manufacturer's instructions. PCR was performed using the ABI 2720 thermal cycler (Applied Biosystems) with the following thermal cycling conditions: 95 °C for 10 min; 50 cycles of 95 °C for 40 s, 62 °C for 30 s, and 60 °C for 2 min followed by 98 °C for 10 min and an indefinite hold at 12 °C. After amplification, fluorescence signals of each droplet were measured by the RainDance Sense instrument according to the manufacturer's instructions. Data were analyzed using the RainDrop Analyst software (RainDance Technologies Inc., version 10.0.7r2).

**HIV-1 near full-length genome sequencing.** Plasma samples from the last lab visit before the initiation of ART were available for 512/1,057 individuals. If it was previously known from Sanger sequencing for genotypic resistance testing that the individual was infected with HIV-1 subtype B, sequencing was done according to the protocol published by Di Giallonardo et al.[61], while in the case of non-B or unknown HIV-1 subtype sequencing was performed according to the protocol published by Gall et al.[62].

**Regression analysis.** For the definition of determinants of HIV-1 reservoir size and long-term dynamics, we used a generalized additive regression analysis using the R package MGCV[30] to include smooth functions of explanatory variables, namely total HIV-DNA at the first time point. As the smoothing function for total HIV-1 DNA at time point 1 we chose a thin plate regression spline with dimension 26 of the basis used to represent the smooth term. Model selection was performed to choose the dimension of the spline by minimizing the Akaike information criterion. Additionally, we used the R package LME4 to perform a mixed-effect model sensitivity analysis[30]. The regression analysis represents a complete case analysis. As a sensitivity analysis, we performed an equivalent regression analysis utilizing multiple imputation for missing values (98 log10 HIV-1 plasma RNA pre-ART values, 34 CD4+ cell count pre-ART values, and 14 individuals had no HIV-1 RNA measurement between 180 days after initiation of ART and the time point of the 1st total HIV-1 DNA measurement) using the R package mice[63]. A sensitivity analysis can be found in the supplementary material (Supplementary Fig. 18). Results of the multivariable analysis remained qualitatively unchanged, apart from (i) the association of nonwhite ethnicity with a slow HIV-1 reservoir decay, for which the p value changed from 0.075 to 0.034 and (ii) the association of female PWID with the decay of the HIV-1 reservoir, which lost significance.

**Explanatory variables.** We defined viral load to be suppressed when all measurements were <50 HIV-1 RNA copies/ml plasma. We defined viral blips to be present when there were measurements ≥50 HIV-1 RNA copies/ml plasma, which were preceded and followed by measurements <50 HIV-1 RNA copies/ml plasma. Any subsequent viral load measurement ≥50 HIV-1 RNA copies/ml plasma within 30 days of a viral blip was considered to be part of the same viral blip[42]. Individuals

who had multiple consecutive viral load measurements ≥50 HIV-1 RNA copies/ml plasma (without experiencing virological failure as defined by two consecutive viral load measurements >200 HIV-1 RNA copies/ml plasma) were considered to exhibit low-level viremia. If an individual exhibited both, viral blips and low-level viremia, we classified this as low-level viremia for our analysis. Sensitivity analysis was performed for these definitions and different viral blips categories (Supplementary Figs. 3 and 4).

We used a hierarchical approach to estimate the HIV-1 infection date on the basis of indicators of varying reliability according to Rusert et al.[57]. The following methods were used with decreasing priority to yield the maximal accuracy for HIV-1 infection dates possible:

1. Defined HIV-1 primary infection: either seroconversion dates (negative and positive HIV-1 screening tests less than 1 year apart) or a diagnosis of a primary infection were available as previously described[64]. We used the midpoint between the dates of the negative and positive tests or, if known, the date of the primary infection as the estimated HIV-1 infection date for these individuals.
2. Defined recent HIV-1 infection: genotypic HIV-1 drug resistance test within the first year after diagnosis revealed low-HIV-1 diversity (less than 0.5% of ambiguous nucleotides) and CD4+ cell counts were >200 cells/μl blood at registration[65–67], we interpreted these as recent HIV-1 infections and used the diagnosis date as an estimate for the infection date.
3. HIV-1 infection date estimates based on a back-calculation method using CD4+ cell counts and their slopes when available[68].
4. For the remaining individuals, no accurate dating was available. For these individuals the date of diagnosis was used as substitute for the HIV-1 infection date, which allowed us to define at least the minimum length of HIV-1 infection.

Time to viral suppression was defined as the time from initiation of ART until the first viral load below 50 copies/ml HIV-1 plasma RNA. If this was not reached, 5.4 years was set as time to viral suppression.

HIV-1 subtype was determined by population sequencing using the REGA HIV-1 subtyping tool in the context of routine genotypic HIV-1 drug resistance testing.

**Reporting summary**. Further information on research design is available in the Nature Research Reporting Summary linked to this article.

## Data availability
The datasets generated during and/or analyzed during the current study are not publicly available due to privacy reasons, the sensitivities associated with HIV infections, and the representativeness of the dataset. However, a coarse-grained version of the dataset is published with the manuscript as Supplementary Data 1.

## Code availability
All code generated during the current study can be made available from the corresponding author on request.

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

## Acknowledgements

The following reagents were obtained through the NIH AIDS Reagent Program, Division of AIDS, NIAID, NIH: Sup-T1 from Dr. Dharam Ablashi; ACH-2 from Dr. Thomas Folks; J-Lat Full Length Cells (clone #10.6) from Dr. Eric Verdin. We thank the patients for participating in the SHCS, the study nurses and physicians for excellent patient care, A. Scherrer, A. Traytel, and S. Wild for excellent data management and D. Perraudin and M. Amstutz for administrative assistance. We thank Melissa Robbiani for having carefully edited the paper. This work was funded within the framework of the Swiss HIV Cohort Study (SNF grant# 33CS30_177499 to H.F.G.). The data were gathered by the Five Swiss University Hospitals, two Cantonal Hospitals, 15 affiliated hospitals and 36 private physicians (listed in http://www.shcs.ch/180-health-care-providers). The work was furthermore supported by the Systems.X grant # 51MRP0_158328 (to N. Beerenwinkel, J. Bogojeska, J.F., V.R., R.D.K., H.F.G. and K.J.M.), by SNF grant 324730B_179571 (to H.F.G.), SNF grant SNF 310030_141067/1 (to H.F.G. and K.J.M.), SNF grants no. PZ00P3-142411 and BSSGI0_155851 to R.D.K., the Yvonne-Jacob Foundation (to H.F.G.), the University of Zurich's Clinical Research Priority Program viral infectious disease, ZPHI (to H.F.G) and the Vontobel Foundation (to H.F.G. and K.J.M.).

## Author contributions

N. Beerenwinkel., J. Bogojeska, J.F., V.R., R.D.K., H.F.G. and K.J.M. conceived and designed the study and analyzed data. C.v.s., V.V., K.N., Y.L.K. and K.J.M. designed and performed the experiments and analyzed the data. N. Bachmann, T.T., C.W.T., A.B., S.P., M.W. and R.D.K. conducted computational analyses and contributed analysis tools and data analysis. J. Böni, M.P., T.K., S.Y., M.B., A.R., M.H., E.B., M.C., H.F.G. and the members of the Swiss HIV Cohort Study conceived and managed the SHCS and ZPHI cohorts collected and contributed patient samples and clinical data. N. Bachmann, R.D.K., H.F.G. and K.J.M. wrote the paper, which all coauthors commented on.

## Additional information

**Competing interests:** M.C. has received the research and travel grants for his institution from ViiV and Gilead. E.B. has received fees for his institution for participation to advisory board from MSD, Gilead Sciences, ViiV Healthcare, Abbvie and Janssen. M.B. has received research or educational grants by Abbvie AG, Gilead Sciences Switzerland Sàrl, Janssen-Cilag AG, MSD Merck Sharp & Dohme AG and ViiV Healthcare GmbH. T. K. has received honoraria from Gilead Sciences and Roche Diagnostics. A.R. has received honoraria for advisory boards and/or travel grants: Janssen-Cilag, MSD, Gilead Sciences, Abbvie, and Pfizer. Unrestricted research grant: Gilead Sciences. All remuneration went to his home institution and not to AR personally. R.D.K. has received grants from the Swiss National Science Foundation and personal fees from Gilead Sciences, outside the submitted work. H.F.G. has received unrestricted research grants from Gilead Sciences and Roche; fees for data and safety monitoring board membership from Merck; consulting/advisory board membership fees from Gilead Sciences, Sandoz and Mepha. K. J.M. has received travel grants and honoraria from Gilead Sciences, Roche Diagnostics, GlaxoSmithKline, Merck Sharp & Dohme, Bristol-Myers Squibb, ViiV and Abbott; and the University of Zurich received research grants from Gilead Science, Roche, and Merck Sharp & Dohme for studies that Dr. Metzner serves as principal investigator, and advisory board honoraria from Gilead Sciences. Remaining authors declare no competing interests.

## the Swiss HIV Cohort Study

Alexia Anagnostopoulos[1], Manuel Battegay[13], Enos Bernasconi[16], Jürg Böni[2], Dominique L. Braun[1], Heiner C. Bucher[18], Alexandra Calmy[12], Matthias Cavassini[17], Angela Ciuffi[19], Günter Dollenmaier[20], Matthias Egger[21], Luigia Elzi[12], Jan Fehr[1], Jacques Fellay[6,7], Hansjakob Furrer[13], Christoph A. Fux[22], Huldrych F. Günthard[1,2], David Haerry[23], Barbara Hasse[1], Hans H. Hirsch[10,12], Matthias Hoffmann[15], Irene Hösli[24], Michael Huber[2], Christian Kahlert[14,25], Laurent Kaiser[11], Olivia Keiser[21], Thomas Klimkait[11], Roger D. Kouyos[1,2], Helen Kovari[1], Bruno Ledergerber[1], Gladys Martinetti[26], Begona Martinez de Tejada[27], Catia Marzolini[12], Karin J. Metzner[1,2], Nicolas Müller[1], Dunja Nicca[14], Paolo Paioni[28], Guiseppe Pantaleo[9], Matthieu Perreau[10], Andri Rauch[14], Christoph Rudin[29], Alexandra U. Scherrer[1,2], Patrick Schmid[14], Roberto Speck[1], Marcel Stöckle[12], Philip Tarr[30], Alexandra Trkola[2], Pietro Vernazza[14], Gilles Wandeler[13], Rainer Weber[1] & Sabine Yerly[12]

[18]Basel Institute for Clinical Epidemiology and Biostatistics, University Hospital Basel, University of Basel, Basel, Switzerland. [19]Institute of Microbiology, University Hospital Lausanne, University of Lausanne, Lausanne, Switzerland. [20]Centre for Laboratory Medicine, Canton St. Gallen, St. Gallen, Switzerland. [21]Institute of Social and Preventive Medicine, University of Bern, Bern, Switzerland. [22]Clinic for Infectious Diseases and Hospital Hygiene, Kantonsspital Aarau, Aarau, Switzerland. [23]Positive Council, Zurich, Switzerland. [24]Clinic for Obstetrics, University Hospital Basel, University of Basel, Basel, Switzerland. [25]Children's Hospital of Eastern Switzerland, St. Gallen, Switzerland. [26]Cantonal Institute of Microbiology, Bellinzona, Bellinzona, Switzerland. [27]Department of Obstetrics and Gynecology, University Hospital Geneva, University of Geneva, Geneva, Switzerland. [28]University Children's Hospital, University of Zurich, Zurich, Switzerland. [29]University Children's Hospital, University of Basel, Basel, Switzerland. [30]Kantonsspital Baselland, University of Basel, Basel, Switzerland

