## [Peer Review File · Nature Communications]

Reviewers' comments:

Reviewer #1 (Remarks to the Author):

This is an interesting, well-written manuscript. Authors evaluate changes in HIV-reservoir size in 1057 individuals on antiretroviral therapy (ART) with suppressed viral load over 10+ years. Reservoir size decreases initially, but appears to plateau after about 6 years. This means that usual ART regimens alone will not eradicate HIV reservoirs, even if viral load is suppressed over many years of ART; an important insight in the search for an HIV cure. Authors also identify several factors associated with slower reservoir decline, such as viral blips, not starting ART within the first year of infection, or high pre-treatment viral load. These factors are biologically plausible. The "reservoir size" was measured as total HIV-1 DNA in longitudinal PBMC samples; this marker appears reasonable. The PBMC samples were obtained at 1.5 years after ART initiation, at 3.5 years, and at 5.5 years; for about half of the participants (n=412), a 4th sample was obtained at > 8 years (median 10 years).

Main strengths:

1. This study includes a large, reasonably diverse, well-characterized cohort of 1057 participants in the Swiss HIV cohort Study. The sample size is more than 10 times larger than any other previous studies that evaluated the decay in HIV reservoirs over time, which contained 30-101 participants. Large sample sizes are needed because of the large between-subject variabilities; this manuscript is a welcome addition to the literature.
2. The study was well-designed. The placement of the data collection times within tight windows around the 1.5, 3.5, 5.5 years resulted in homogeneous data collection, and allowed the use of straightforward, robust statistical methods to estimate the rate of decline in the HIV reservoir. Statistical analyses were done carefully. Data presentations are comprehensive, with extensive supporting information in the Supplement, and well-designed figures. As an example for the careful analyses: Identifying factors associated with the size of the reservoir at 1.5 years and the slope of the reservoir decline is difficult because factors are correlated; separating the impact of HIV-1 subtype, race/ethnicity and HIV transmission group as predictors was well-done.
3. The presented results are plausible.
4. The manuscript overall is well-written.

Main concerns:

1. Lines 152-155: Authors used linear regression to estimate the slope of HIV reservoir decline from year 1 through 5.5 for the entire cohort of 1057 participants (correct). They then estimated the linear regression slope from year 1 through year 12 based on the 412 participants with median follow-up of 10 years, and used this to calculate an estimated half-life of 11.2 years. However, it is obvious from Figures 2a and 2c that the HIV reservoir plateaus after about 6 years; the median at year 1 is 2.75 (log₁₀ HIV-1 DNA copies/mge), the median at year 6 and beyond is around 2.5 – the 50% point of 2.45 copies/mge is never reached. If authors want to present the "estimated half-life" they should explicitly point out that this is a mathematical exercise, and does not reflect the observed data -- the median reservoir size gets reduced by less than 50% even over 10+ years.
2. Lines 263-264: "Low viral loads pre-ART showed a trend towards slower reservoir decay." Figure 3b shows no such trend.
3. Discussion, lines 334-5: "We found a ... continuous decay of the ... reservoir from years 1.5 to 10 ..., which slowed down over time" is misleading. The data showed a plateau after year 6, which is qualitatively different.
4. Discussion, lines 336-7 and 388-391: "26.6% of individuals exhibited a positive slope of total HIV-1 DNA levels". A) This number was for the slope from year 1.5 to year 5.5; the wording here suggests that the positive slope was for years 1.5-10. B) There is a large between-subject variability in slopes; a good deal of that may be due to assay variability or within-subject sample variability – slope estimates based on 3 data points simply are not that stable. A substantial part of the positive slope estimates may be due to assay and sample variability, and does not necessarily reflect increasing reservoir size (if we could measure reservoirs reliably). This should be mentioned in the discussion.

5. Discussion, line 352-3 and following discussion of reservoir size as cause for viral blips: Authors argue that lack of adherence can be ruled out as a major factor for viral blips. I disagree. Self-reported adherence is not a perfect measure of adherence. And the argument that HIV reservoir size at 1.5 years is predictive of the blips (and slower decline) does not imply that larger HIV reservoirs cause viral blips – the causation may well be the other way around. Similarly, the fewer viral blips among those who were treated within the first year of HIV infection may reflect that patients who get an early diagnosis may be more health-conscious and have better adherence and thus fewer blips. The conclusions should be toned down here.

Other concerns (easily fixable):

1. Lines 150-151: Authors used linear regression to estimate the slope of HIV reservoir decline from year 1 through 5.5 for the entire cohort of 1057 participants (correct). They then calculated the half-life of the reservoir as 5.6 years based on the slope of -0.054. This is correct if the half-life is calculated as the 50% reduction point of the HIV reservoir on the original scale (DNA copies/mge), not on the log₁₀ scale. Since the slope and median DNA levels are all given on the log₁₀ scale, authors should point out that the half-life is calculated for the 50% reduction on the original scale.
2. Figures 2a and 2c: The size of the HIV-1 reservoir through years 1.5 – 5.5 is based on n=1057 participants, while the last time point at median 10 years is available only for n=412 participants. While the sample sizes are clearly demarked on the figures, there is the possibility of a cohort effect. It would be desirable to also see an analysis of the HIV reservoir size restricted to the 412 participants with long-term follow-up, as part of the supplemental materials. This would also strengthen the argument of the plateau after year 6.
3. Lines 278-282: “Interferon treatment of HCV infection showed a trend towards triggering a faster decay of the HIV-1 reservoir. ... (p=0.19999)”. This is based on 11 individuals. Authors argue that the effect was not significant because the sample size was small. This is wrong; it is well possible that there is simply no signal. “Trends” with a p-value of 0.20 may be spurious findings. More generally, it would be sensible to NOT report any associations with p-values > 0.05, given the overall large sample size, but also given the multiple comparisons in this analysis.
4. Lines 293-4: Instead of “slope is 1.79 fold ... lower”, it would be better to say “... 1.79-fold less steep”. Also, the slope was calculated on the log₁₀-scale of HIV-1 DNA – this needs to be expressed correctly, the manuscript cites it as “DNA slope”, which implies the original scale. Same for the label of the y-axes in Figure 4 a-b.
5. Figure 4c is confusing. Could delete.
6. Lines 410-411: The final conclusion is rather weak, “This suggests that viral blips and low-level viremia are relevant parameters to monitor in future reservoir and cure studies.” An important result of this study is that the viral DNA reservoir plateaus after 6 years of suppressive ART, which also has consequences for cure studies. Consider commenting.
7. Figure S7 caption: reference to the outcomes in the first sentence is confusing. What are DNA1, DNA2, DNA3?

Reviewer #2 (Remarks to the Author):

The authors describe a study of a large cohort of 1,057 HIV-infected individuals on suppressive antiretroviral therapy (ART) for a median of 5.4 years, assessing viral and host characteristics that associate with reservoir size and long-term dynamics. Eligible individuals had to be on a potent ART regimen, have had no interruption of treatment for more than a week, no virologic failure (2 consecutive VLs > 200 cpm), and have sufficient cell samples available for quantitating HIV reservoir size (3-4 longitudinal time points post start of ART). The overall subset of patients was very diverse, representative of the real-world setting, which was dominated by white ethnicity, male gender, male-male transmission and infection with subtype B. However, sufficient women, non-white ethnicities and individuals with non-B subtype infection were included to allow meaningful comparisons.

HIV-1 reservoir size was measured by droplet digital PCR (ddPCR) assay as total cell-associated HIV-1 DNA in peripheral blood mononuclear cell (PBMC). Viral blips were defined as plasma viral loads [VLs] ≥ 50 cpm with values <50 cpm before and after, unless within 30 days of blip – then considered as part of the viral blip), while low-level viremia was defined as multiple consecutive VLs ≥ 50 cpm without virologic failure of 2x consecutive >200 cpm). These were the key virologic measures utilized to investigate the effects of many different parameters (viral and host) on HIV-1 reservoir size and long-term dynamics.

A relationship between earlier initiation of ART and smaller HIV-1 reservoir, as shown in many studies, was confirmed. HIV-1 reservoir size at 1.5 years after ART initiation predicted VL blips before and after the start of ART, and associated with a slower decay of the viral reservoir. These findings suggest the measure of viral blips may be biologically meaningful, somehow reflecting underlying replicative activity even if this may reflect what is occurring in tissue sites and not necessarily circulating reservoir cells. Studies have shown that low levels of virus produced with suppressive ART in some patients, are invariant clones of virus that are rarely represented in the peripheral HIV DNA reservoir – implicating cell sources other than circulating CD4 T cells (e.g. Bailey et al, 2006 – which has been referenced) – this has important implications for how we think about the respective virus/provirus measures, HIV-1 persistence and virologic failure on ART. Total HIV-1 and residual virus are distinct measures, and HIV-1 DNA levels do not predict levels of residual viremia. The data from the current study support a complex interplay between residual virus and HIV-1 reservoir size and long-term dynamics.

Strengths of the study: The Swiss HIV Cohort Study (SHCS) – a long standing (since 1988) observational study which enrolls HIV-infected adults with all modes of transmission, provides an amazing resource of samples and data collected over time. The majority of ART-treated patients ($>75\%$) in Switzerland are represented in this cohort – showing that the subset of patients studied are very representative of the ART-treated HIV-infected population. Few studies have directly asked the question of HIV-1 reservoir decay during ART and how this might relate to residual viremia; those that have were conducted on substantially smaller numbers of patients compared to the current study. Limitations of the study have been stated and addressed well in the discussion. The measure of total HIV-1 DNA has stood the test of time and provides a good proxy for HIV reservoir size – despite over-estimation in terms of the actual replication-competent viral genomes present. Certainly, in the context of large cohort studies or clinical trials it serves as a robust and practical measure that gives insights into the HIV reservoir that is different from the measure of residual VLs.

Specific comments:

1. It is particularly interesting that non-white ethnicity associated with lower HIV-1 reservoir size at 1.5 years after ART initiation. It would be helpful to have a breakdown of the non-white group of individuals – from where in Africa/elsewhere do they come from? As host genetic variability is most extensive in Africa - it would be important that this is at least documented, as may be important for future studies.

2. That non-B subtypes had lower HIV-1 DNA levels compared to HIV-1 subtype B in the white ethnic group – suggests viral subtype differences may be an important determinant in HIV-1 reservoir size. This needs some discussion, are there data from studies that have directly looked at disease progression with different HIV-1 subtypes in the same population that could support this finding.

3. Did the authors directly test whether there are differences in HIV-1 reservoir measures between men and women? It has been shown that women have better control of HIV-1 in early untreated infection (lower VLs) but in chronic infection progress more quickly compared to men – so in the long term progress to AIDS in similar time. This would suggest that perhaps women might have an additional advantage in achieving a low HIV-1 reservoir if treated early compared to men.

4. The smaller reservoir size in the PWID transmission group after correction for IFN treatment for HCV – how can one explain this finding? Not sure if there are any data on other modes of transmission e.g. blood transfusion – where, as with PWIDs, one would expect infection occurs with a quasispecies of viral variants. This is very different to mucosal transmission - where a bottleneck occurs with

transfer of one or very few viral variants with other modes of transmission (sexual, mother-to-child transmission).

5. Another intriguing finding is the subset of individuals (26.6%) who did not achieve a negative slope of their total HIV-1 DNA levels after ART initiation. This finding warrants more discussion – what were the characteristics of this specific group, does anything stand out, what could explain this outcome for a substantial proportion of the cohort?

Caroline T. Tiemessen

Reviewer #3 (Remarks to the Author):

Review "DETERMINANTS OF HIV-1 RESERVOIR SIZE AND LONG-TERM DYNAMICS DURING SUPPRESSIVE ART"

General

The authors have explored associations of the HIV-1 reservoir size and long-term decay with various host and viral factors in a large scale observational study. This complex interplay is shown in an in-depth analysis in an impressive patient cohort (n= 1057) and followed for an extensive time with HIV-1 DNA measured at 3 timepoints: 1.5 years, 3.5 years and 5.4 years post-ART initiation (n=645) and in some patients (n=412) 4th timepoint was available at 10 years after ART was initiated. This study provides a comprehensive analysis of determinants of the HIV-1 reservoir size and dynamics in patients on suppressive ART. It is unique in such a large scale. However, no new conclusions are drawn in this study, neither none of the proposed biological mechanisms explaining interplay between the residual viremia and HIV-1 DNA reservoir are being explored. Thus, this is purely a descriptive study with the most prominent strength of a well characterized cohort of an impressive number of individuals included.

Most important findings of the study are the shown associations between viral blips and low-level viremia with slower reservoir decay, timing of ART initiation, pre-ART viral load. Longer time on ART was shown to be a predictor for a smaller reservoir. Furthermore, significant decrease in the size of HIV-1 DNA reservoir was shown 1.5 to 5.4 years post-ART, which plateaus afterwards. And interestingly, non-white ethnicity seemed to be associated with a smaller reservoir size.

Major comments

Lack of novelty, purely observational study. The size of the analyzed patient group and the number of samples analyzed per patient is impressive and a study of this magnitude is able to draw conclusions with a high statistical power.

Lines 216-219: Showing smaller reservoir size associated with HIV-1 non-B subtype, where indeed evaluation of the PCR assay needs to be performed. How were the HIV-1 full-length sequences obtained? This would need to be described better in the manuscript. Here, an in silico analysis is mentioned, where no PCR mismatches were detected in 568/1368 individuals. This needs a better elaboration. Which patients are representing these 568 patients?

In Figure 4, authors are illustrating the complex interplay of residual viremia and HIV-1 reservoir size and its long-term dynamics, with two possible scenarios explaining this interplay and long-term dynamics. However, authors do not bring here an analysis studying these proposed mechanisms. The study remains purely descriptive. Adding a more mechanistic exploration of the HIV-1 reservoir would enhance this study substantially. Would that be possible with the samples collected?

In 1057 of 1166 patients, total HIV-1 DNA quantification was achieved. What was the cause why it did not work in the other samples? Low quality of samples? Or were these all samples with a low HIV-1 DNA level (not quantifiable), so are patients excluded with a very low HIV-1 DNA reservoir? (biased towards patients that started ART very early?)

Is there more information available about the Fiebig stages of the patients that initiated ART within 1 year? This could give interesting additional data.

Minor comments

An interesting finding mentioned is the significant effect of non-white ethnicity, however, it is not clearly mentioned whether possible confounding effect may be biasing this finding.

Figure 1b: The figure can be misleading: it insinuates that there is a blip around every HIV-1 DNA quantification.

What is the range of copies/ml of the blips? (Maximum level?)

PWID: abbreviation is not explained in the text, only in the figure.

Figure 4c: This figure is referred to only in lines 299-300. The figure could use a more detailed explanation in the text and make the link more comprehensible with the text.

The clinical relevance of 1.93 and 1.79-fold difference at slope values of $-0.06 \rightarrow -0.04$ should be more elaborated.

Point by point reply

We are grateful for the reviewers very favorable assessment of our work and deeply appreciate their thoughtful comments and constructive suggestions, which we have fully addressed in this revision and believe that this has strengthened the manuscript significantly. Our point by point reply contains reviewer's comments in blue, answers in black, and changes in the manuscript in brown.

Reviewer #1:

This is an interesting, well-written manuscript. Authors evaluate changes in HIV-reservoir size in 1057 individuals on antiretroviral therapy (ART) with suppressed viral load over 10+ years. Reservoir size decreases initially, but appears to plateau after about 6 years. This means that usual ART regimens alone will not eradicate HIV reservoirs, even if viral load is suppressed over many years of ART; an important insight in the search for an HIV cure. Authors also identify several factors associated with slower reservoir decline, such as viral blips, not starting ART within the first year of infection, or high pre-treatment viral load. These factors are biologically plausible. The "reservoir size" was measured as total HIV-1 DNA in longitudinal PBMC samples; this marker appears reasonable. The PBMC samples were obtained at 1.5 years after ART initiation, at 3.5 years, and at 5.5 years; for about half of the participants (n=412), a 4th sample was obtained at > 8 years (median 10 years).

Main strengths:

- 1. This study includes a large, reasonably diverse, well-characterized cohort of 1057 participants in the Swiss HIV cohort Study. The sample size is more than 10 times larger than any other previous studies that evaluated the decay in HIV reservoirs over time, which contained 30-101 participants. Large sample sizes are needed because of the large between-subject variabilities; this manuscript is a welcome addition to the literature.*
- 2. The study was well-designed. The placement of the data collection times within tight windows around the 1.5, 3.5, 5.5 years resulted in homogeneous data collection, and allowed the use of straightforward, robust statistical methods to estimate the rate of decline in the HIV reservoir. Statistical analyses were done carefully. Data presentations are comprehensive, with extensive supporting information in the Supplement, and well-designed figures. As an example for the careful analyses: Identifying factors associated with the size of the reservoir at 1.5 years and the slope of the reservoir decline is difficult because factors are correlated; separating the impact of HIV-1 subtype, race/ethnicity and HIV transmission group as predictors was well-done.*
- 3. The presented results are plausible.*
- 4. The manuscript overall is well-written.*

Reply: We thank the reviewer for this very positive assessment of our work.

Main concerns:

1. Lines 152-155: Authors used linear regression to estimate the slope of HIV reservoir decline from year 1 through 5.5 for the entire cohort of 1057 participants (correct). They then estimated the linear regression slope from year 1 through year 12 based on the 412 participants with median follow-up of 10 years, and used this to calculate an estimated half-life of 11.2 years. However, it is obvious from Figures 2a and 2c that the HIV reservoir plateaus after about 6 years; the median at year 1 is 2.75 (log₁₀ HIV-1 DNA copies/mge), the median at year 6 and beyond is around 2.5 – the 50% point of 2.45 copies/mge is never reached. If authors want to present the “estimated half-life” they should explicitly point out that this is a mathematical exercise, and does not reflect the observed data -- the median reservoir size gets reduced by less than 50% even over 10+ years.

Reply: We thank the reviewer for pointing this out. Half-lives were used as a measure to quantify the speed of the total HIV-1 DNA decay (without implying that total HIV-1 DNA reaches 50% of the initial value over the observation period). We chose half-lives (rather than for example rates of decay) as a measure for the decay of total HIV-1 DNA, to be consistent with the literature (e.g., (Siliciano et al., 2003)), and thus, ensure comparability. Indeed, the plateauing observed on a log₁₀ scale in **Fig. 2a** implies that the 50% point is not reached in the half-life period. In more technical terms, the assumption of first-order decay kinetics for the half-life estimation is imperfect. We now clarified this by the following changes in the results section:

- We acknowledged that the 50% point was not reached within the half-life period (line 139-141):
“However, the reservoir did not reach 50% of its size at the first measurement within the half-life period, since the HIV-1 reservoir decay slowed down over time.”
- We clarified that we made an assumption by adding the following in lines 138-9:
“..., which corresponds to a median half-life of 5.6 years on the linear scale (total HIV-1 DNA copies/mge) assuming first order decay kinetics.”
- Accordingly, we adjusted the sentence in lines 148-9:
“..., which corresponds to a median half-life of 11.8 years on the linear scale (total HIV-1 DNA copies/mge) assuming first-order decay kinetics.”
- We also implemented this in the discussion, line 368, together with additional changes in regard to our reply to your main concern 4 (please see below).

2. Lines 263-264: “Low viral loads pre-ART showed a trend towards slower reservoir decay.” Figure 3b shows no such trend.

Reply: We agree that this presumable trend was overstated and deleted this part in the sentence (lines 238-9 in the revised manuscript).

3. Discussion, lines 334-5: “We found a ... continuous decay of the ... reservoir from years 1.5 to 10 ..., which slowed down over time” is misleading. The data showed a plateau after year 6, which is qualitatively different.

Reply: Indeed, the total HIV-1 DNA trajectories appear to exhibit a plateau in Fig 2a/c. However, among the individuals with a third and fourth total HIV-1 DNA measurement available, there were still significant differences between the third and fourth HIV-1 DNA

measurement (using paired Wilcoxon test) (**Fig 2a**: p-value of 0.0002 between 5.4 and 10 years after initiation of ART), which indicates that the plateau is not reached in this time. Thus, a decay that slowed down over time and the approach of a plateau are not mutually exclusive. We incorporated the idea of a plateau in the discussion in lines 294-5:

“We found a small but continuous decay of the HIV-1 reservoir from years 1.5 to 10 after initiation of suppressive ART, which slowed down over time and seems to approach a plateau”. To be consistent with what we discussed above, we replaced the term “which plateaus thereafter” in line 152 by “which approaches a plateau thereafter”.

4. Discussion, lines 336-7 and 388-391: “26.6% of individuals exhibited a positive slope of total HIV-1 DNA levels”. A) This number was for the slope from year 1.5 to year 5.5; the wording here suggests that the positive slope was for years 1.5-10.

B) There is a large between-subject variability in slopes; a good deal of that may be due to assay variability or within-subject sample variability – slope estimates based on 3 data points simply are not that stable. A substantial part of the positive slope estimates may be due to assay and sample variability, and does not necessarily reflect increasing reservoir size (if we could measure reservoirs reliably). This should be mentioned in the discussion.

Reply to A: We agree that our previous formulation might be slightly misleading. We clarified this by ...

- ... pointing out that the fraction of individuals with a positive slope was similar in the 412 individuals receiving ART for approximately 10 years. Thus, we added the following sentence to line 149:
“Again, in 100 individuals (24.3%) with four measurements available the total HIV-1 DNA level increased (**Supplementary Fig. S2c**).” (regarding the new **Supplementary Fig. S2** please see also our reply to your other concern 2.)
- ... adapting the sentence under consideration, line 296, by adding:
“Interestingly, 26.6% of those individuals exhibited a positive slope of total HIV-1 DNA levels over a median of 5.4 years of suppressive ART.”
- ... adapting the second sentence under consideration, lines 358-9, by adding:
“Notably, total HIV-1 DNA levels did not decrease after initiation of ART in 281/1,057 (26.6%) individuals over a median of 5.4 years of suppressive ART.”

Reply to B: We acknowledge the reviewers’ concern and now discuss this (see below). However, while assay variability or within-subject sample variability may affect our results in principle, we do not find evidence that those caused a strong bias. In particular, the following reasons argue against assay variability or within-subject sample variability as the main explanation for the high frequency (26.6%) of positive slopes: Firstly, all 3-4 samples of each individual were processed and analyzed in the same droplet digital PCR run (as described in the Methods section). This implies that potential assay variability would affect all samples of each individual to the same extent and could thus not lead to false positive slope estimates. Secondly, we found that in 66.7% (88/132) of cases a positive increase between the first and second total HIV-1 DNA measurement was followed by an increase between the second and third total HIV-1 DNA measurement. If the positive slopes were due to within-subject sample variability, a lower fraction of consistent increase would have been observed (particularly taking into account that overall only a minority of 26.6% exhibits an increasing slope). Thirdly, potential assay variability cannot play a major role because we have confirmed with our

analysis previously described phenomena like an association of low total DNA levels with shorter duration of treatment, and in particular with Fiebig stages (see our reply to reviewer 3, new **Supplementary Fig. S5**), high CD4+ cell count levels pre-ART, etc. These confirmations illustrate that our assay is robust and results are valid. Fourthly, we added a new logistic regression analysis with positive/negative slope as outcome variable (new **Supplementary Fig. S12**). This analysis additionally supports that positive slopes were not due to assay variability or within-subject sample variability. Please also see our response to comment 5 of reviewer 2 below. We made the following revisions in our manuscript:

- We added the following sentence in our results section, line 142:
“Of note, in 281 individuals (26.6%) the total HIV-1 DNA level increased (**Fig. 2b**). Further, we found that in 66.7% (88/132) of cases a positive increase between the first and second total HIV-1 DNA measurement was followed by an increase between the second and third total HIV-1 DNA measurement.”
- We revised the following paragraph in the discussion, line 368-379:
“We modelled the HIV-1 reservoir decay as linear on the log₁₀ scale based on three time points. Together with potential assay and within-subject sample variability, this indicates some uncertainty around individual slopes. Despite these and other potential limitations (e.g., not differentiating between replication competent- and -defective viruses), our study shows that total HIV-1 DNA measured in PBMC is a robust proxy for the latent HIV-1 reservoir size after the first rapid decay following initiation of ART for several reasons. It correlates independently with time to initiation of ART, with CD4+ cell count, and with viral blips. In addition, the decay of the latent HIV-1 reservoir is in line with smaller studies using either viral outgrowth assays or total HIV-1 DNA^{17,19}. Furthermore, if slopes were due to within-subject sample variability, a lower fraction of consistent within-subject increase/decrease between samples would have been observed.”

5. Discussion, line 352-3 and following discussion of reservoir size as cause for viral blips: Authors argue that lack of adherence can be ruled out as a major factor for viral blips. I disagree. Self-reported adherence is not a perfect measure of adherence. And the argument that HIV reservoir size at 1.5 years is predictive of the blips (and slower decline) does not imply that larger HIV reservoirs cause viral blips – the causation may well be the other way around. Similarly, the fewer viral blips among those who were treated within the first year of HIV infection may reflect that patients who get an early diagnosis may be more health-conscious and have better adherence and thus fewer blips. The conclusions should be toned down here.

Reply: We agree that self-reported adherence in general might not be a perfect measure of adherence. However, it is probably the only way one can measure adherence systematically in thousands of patients over decades in a routine clinical setting. In the light of those concerns, the SHCS has investigated the value of self-reported adherence in the cohort in several independent studies showing that self-reported adherence predicts several key clinical outcomes: For instance, it was shown that self-reported adherence was predictive not only for virological failure but also for death (Glass et al., 2015). The prediction of those two crucial clinical endpoints by the self-reported adherence measure clearly shows the validity of this proxy of adherence in the set-up of the SHCS. Furthermore, the validity of self-reported adherence is supported by another SHCS study finding of a significant linear trend in optimal

viral suppression by self-reported missed doses and an association of nonadherence with younger age, boosted protease inhibitor regimens, and other factors (Glass et al., 2006).

In this context it is crucial to consider the effect size of blips on the HIV-1 reservoir decay, with or without including adherence. Most importantly, we found that upon adjusting with our proxy of adherence, the effect-sizes for blips and low-level viremia changed only marginally: The effect size of blips on a slow decay decreased from 0.0277 to 0.0267 (3.7%) while the effect size of low-level viremia decreased from 0.0313 to 0.0309 (1.3%) when we corrected for adherence (lines 262-3; **Supplementary Fig. S16**). This indicates that it is highly unlikely that the effect of blips and low-level viremia is due to confounding by adherence. This furthermore indicates, that even adjusting with other potentially more precise measures of adherence, for instance, observed medication adherence, would not have substantially changed the effect size of blips/low level viremia, because if that had been the case, the validated proxy used here should have had a stronger effect.

Finally, for the analysis of the size of the HIV-1 reservoir we only included information on self-reported adherence before that sample was taken, while for the analysis of the decay of the HIV-1 reservoir we only included information on self-reported adherence between the first and third sample used to calculate the decay. Thus, the observations are distinctly separated by time, which makes a reversed causality highly unlikely.

One possible reason for the neglectable impact of adherence in our study is that the individuals enrolled are highly selected for optimal adherence, because these are patients who did not experience treatment failure for a median of 5.4 years. On that note, 87% of our study population never reported to have missed more than 1 pill in the previous month at all bi-annual SHCS follow-up visits between the first and third sample (this information was available for 1051/1057 individuals).

To emphasize that self-reported adherence may be a limitation in principle but also why we think that this limitation does not affect the main conclusion of our study, we added the following sentences in the discussion, lines 314-320:

“...self-reported adherence was high in the study population, i.e., 920/1,051 (87%) individuals never reported to have missed more than 1 pill in the previous month at all bi-annual SHCS follow-up visits between the first and third time point. Despite being self-reported (and hence subject to potential biases), the adherence measure reported in the SHCS is a validated prediction marker for important clinical outcomes such as virological failure and mortality (Glass et al., 2015). Furthermore, we found that upon adjusting with the proxy of adherence, the effect-sizes for blips and low-level viremia changed only marginally.”

Other concerns (easily fixable):

1. Lines 150-151: Authors used linear regression to estimate the slope of HIV reservoir decline from year 1 through 5.5 for the entire cohort of 1057 participants (correct). They then calculated the half-life of the reservoir as 5.6 years based on the slope of -0.054. This is correct if the half-life is calculated as the 50% reduction point of the HIV reservoir on the original scale (DNA copies/mge), not on the log10 scale. Since the slope and median DNA levels are all given on the log10 scale, authors should point out that the half-life is calculated for the 50% reduction on the original scale.

Reply: This is a valid point that we clarified in the manuscript as follows:

- We added in line 138:

“...., which corresponds to a median half-life of 5.6 years on the linear scale (total HIV-1 DNA copies/mge) assuming first order decay kinetics.”

- Accordingly, in line 148:

“... , which corresponds to a median half-life of 11.8 years on the linear scale (total HIV-1 DNA copies/mge) assuming first-order decay kinetics.”

2. Figures 2a and 2c: The size of the HIV-1 reservoir through years 1.5 – 5.5 is based on n=1057 participants, while the last time point at median 10 years is available only for n=412 participants. While the sample sizes are clearly demarked on the figures, there is the possibility of a cohort effect. It would be desirable to also see an analysis of the HIV reservoir size restricted to the 412 participants with long-term follow-up, as part of the supplemental materials. This would also strengthen the argument of the plateau after year 6.

Reply: We thank the reviewer for the valuable feedback and agree that a possible cohort effect merits further elaboration. We now included a new **Supplementary Fig. S2** in the Supplementary Material (see below), which corresponds to **Fig. 2** in the main manuscript restricted to the 412 patients with a fourth sample available. Since the same patterns as in the analysis including 1,057 individuals were observed, we can exclude the possibility of a cohort effect. The result section was modified as follows, lines 130-5:

“On the population level, log₁₀ total HIV-1 DNA levels significantly decreased with diminishing differences over time to 2.59 (IQR: 2.30-2.85; n=1,057), 2.53 (IQR: 2.23-2.77; n=1,057), and 2.52 (IQR: 2.22-2.7; n=412) median log₁₀ total HIV-1 DNA copies/mge at 3.5, 5.4, and 10.0 years after initiation of ART, respectively (**Fig. 2a, 2c**). A subgroup analysis of the 412 individuals with four available total HIV-1 DNA quantifications yielded qualitatively equivalent results excluding the possibility of a cohort effect (**Supplementary Fig. S2**).”

Figure S2. The HIV-1 reservoir size and long-term dynamics in 412 individuals on suppressive ART for on median 10 years. (a) Beanplot of total HIV-1 DNA levels in 412

individuals on long-term suppressive ART at 4 different time points (with median 1.5, 3.5, 5.5, and 10.0 years after initiation of ART) and the respective sample size. The p-values were calculated using paired Wilcoxon tests. The individual observations are shown as small lines (gray or black) in a one-dimensional scatter plot. Overlaid is the estimated density of the distributions (filled in pink) and the median is depicted by a black line. (b) Histogram of linear regression slope over the four measurements of total HIV-1 DNA levels with median 1.5-10 years after initiation of ART. (c) Spline fitted to all log₁₀ total HIV-1 DNA/1 million genomic equivalents showing the 95% confidence intervals in blue and sampling times after initiation of ART in years on the x-axis. ART, antiretroviral therapy.

3. Lines 278-282: “Interferon treatment of HCV infection showed a trend towards triggering a faster decay of the HIV-1 reservoir. ... (p=0.19999)”. This is based on 11 individuals. Authors argue that the effect was not significant because the sample size was small. This is wrong; it is well possible that there is simply no signal. “Trends” with a p-value of 0.20 may be spurious findings. More generally, it would be sensible to NOT report any associations with p-values > 0.05, given the overall large sample size, but also given the multiple comparisons in this analysis.

Reply: We thank the reviewer for this suggestion. This sentence has been removed (lines 255-57 in the revised manuscript).

4. Lines 293-4: Instead of “slope is 1.79 fold ... lower”, it would be better to say “... 1.79-fold less steep”. Also, the slope was calculated on the log₁₀-scale of HIV-1 DNA – this needs to be expressed correctly, the manuscript cites it as “DNA slope”, which implies the original scale. Same for the label of the y-axes in Figure 4 a-b.

Reply: We adjusted the sentence in line 270:

“The predicted log₁₀ total HIV-1 DNA slope is 1.79-fold and 1.93-fold less steep in individuals with viral blips and low-level viremia, ...”

Furthermore, the labels of the y-axes in **Figs. 4a and 4b** have been changed to “predicted log₁₀ total HIV-1 DNA slope”. In accordance, we changed “predicted total HIV-1 DNA slope” to “predicted log₁₀ total HIV-1 DNA slope” in lines 269 and 274 in the text and in lines 864 and 867 in the figure legend.

5. Figure 4c is confusing. Could delete.

Reply: We agree with the reviewer that the explanations concerning **Fig. 4c** were insufficient. However, we believe that the figure is valuable to readers as it captures the concept of the underlying study. Thus, we improved its comprehensibility at several levels:

- Within the figure, we replaced “*corrected for the initial size” by “*corrected for the HIV-1 reservoir size 1.5 years after the initiation of ART” to use the same terminology as in the text.
- We adjusted the figure legend, lines 874-880, by adding a detailed explanation of the concept captured by the figure:

“(c) Conceptual figure showing the observed associations between: (i) residual viremia, (ii) the HIV-1 reservoir size 1.5 years after initiation of ART, and (iii) the decay of the HIV-1 reservoir 1.5 to 5.4 years after initiation of ART. Residual viremia captured both, low-level viremia and viral blips, from 1.5 to 5.4 years after initiation of ART and

was thus indicated as the mean log₁₀ HIV-1 RNA. Arrows indicate a positive effect size, i.e., an enhancing effect. The blunted arrow indicates an inhibiting effect. P-values were derived using linear regression.”

- We added the following sentence in the results section, line 282:
“The HIV-1 reservoir size is associated with both, subsequent residual viremia and the subsequent HIV-1 reservoir decay, while simultaneously residual viremia inhibits the HIV-1 reservoir decay (Fig. 4c).”

6. Lines 410-411: The final conclusion is rather weak, “This suggests that viral blips and low-level viremia are relevant parameters to monitor in future reservoir and cure studies.” An important result of this study is that the viral DNA reservoir plateaus after 6 years of suppressive ART, which also has consequences for cure studies. Consider commenting.

Reply: Currently, viral blips are clinically often ignored, respectively found to be of no clinical importance. Our result that viral blips and the size of the HIV-1 reservoir are significantly correlated is a novel important finding of this study. Thus, for future cure studies it will be beneficial to include first patients not showing any viral blips or low-level viremia. Residual replication, which could be one explanation for the observed viral blips, might mask potential effects of novel cure strategies. In theory, if no differentiation between patients with and without viral blips is made, results of cure studies could be falsely negative because of confounding by residual replication. Such a scenario could easily be possible because current cure studies are mostly small pilot studies. Our study, however, did not allow to prove causality between residual viremia with HIV-1 reservoir size and long-term dynamics, which we stated in our discussion. Thus, we think that the association between viral blips and total HIV-1 DNA is an important finding. Nevertheless, we agree with the reviewer that it would be good to add our finding of the decelerating HIV-1 DNA decay in the final conclusion, line 386:

“The very slow and decelerating HIV-1 DNA decay after on median 5.4 years also reiterates the need for targeted new interventions to reduce the HIV-1 reservoir, because long-term ART seems not to substantially affect it.”

7. Figure S7 caption: reference to the outcomes in the first sentence is confusing. What are DNA1, DNA2, DNA3?

Reply: We thank the reviewer for pointing out this lack of clarity. We adjusted the figure legend in lines 163-166 of the Supplementary Material:

“Determinants of HIV-1 reservoir long-term dynamics analyzed with linear mixed effect model (implemented in the R package “lme4”) and differences of total HIV-1 DNA levels compared to the first total HIV-1 DNA measurement as outcome (i.e., zero for the 1st time point, difference of 1st and 2nd measurement for the second time point, and difference of 1st and 3rd measurement for the third time point).”

Reviewer #2:

The authors describe a study of a large cohort of 1,057 HIV-infected individuals on suppressive antiretroviral therapy (ART) for a median of 5.4 years, assessing viral and host characteristics that associate with reservoir size and long-term dynamics. Eligible individuals had to be on a potent ART regimen, have had no interruption of treatment for more than a

week, no virologic failure (2 consecutive VLs > 200 cpm), and have sufficient cell samples available for quantitating HIV reservoir size (3-4 longitudinal time points post start of ART). The overall subset of patients was very diverse, representative of the real-world setting, which was dominated by white ethnicity, male gender, male-male transmission and infection with subtype B. However, sufficient women, non-white ethnicities and individuals with non-B subtype infection were included to allow meaningful comparisons. HIV-1 reservoir size was measured by droplet digital PCR (ddPCR) assay as total cell-associated HIV-1 DNA in peripheral blood mononuclear cell (PBMC). Viral blips were defined as plasma viral loads [VLs] ≥ 50 cpm with values <50 cpm before and after, unless within 30 days of blip – then considered as part of the viral blip), while low-level viremia was defined as multiple consecutive VLs ≥ 50 cpm without virologic failure of 2x consecutive >200 cpm). These were the key virologic measures utilized to investigate the effects of many different parameters (viral and host) on HIV-1 reservoir size and long-term dynamics. A relationship between earlier initiation of ART and smaller HIV-1 reservoir, as shown in many studies, was confirmed. HIV-1 reservoir size at 1.5 years after ART initiation predicted VL blips before and after the start of ART, and associated with a slower decay of the viral reservoir. These findings suggest the measure of viral blips may be biologically meaningful, somehow reflecting underlying replicative activity even if this may reflect what is occurring in tissue sites and not necessarily circulating reservoir cells. Studies have shown that low levels of virus produced with suppressive ART in some patients, are invariant clones of virus that are rarely represented in the peripheral HIV DNA reservoir – implicating cell sources other than circulating CD4 T cells (e.g. Bailey et al, 2006 – which has been referenced) – this has important implications for how we think about the respective virus/provirus measures, HIV-1 persistence and virologic failure on ART. Total HIV-1 and residual virus are distinct measures, and HIV-1 DNA levels do not predict levels of residual viremia. The data from the current study support a complex interplay between residual virus and HIV-1 reservoir size and long-term dynamics.

Strengths of the study: The Swiss HIV Cohort Study (SHCS) – a long standing (since 1988) observational study which enrolls HIV-infected adults with all modes of transmission, provides an amazing resource of samples and data collected over time. The majority of ART-treated patients (>75%) in Switzerland are represented in this cohort – showing that the subset of patients studied are very representative of the ART-treated HIV-infected population. Few studies have directly asked the question of HIV-1 reservoir decay during ART and how this might relate to residual viremia; those that have were conducted on substantially smaller numbers of patients compared to the current study.

Limitations of the study have been stated and addressed well in the discussion. The measure of total HIV-1 DNA has stood the test of time and provides a good proxy for HIV reservoir size – despite over-estimation in terms of the actual replication-competent viral genomes present. Certainly, in the context of large cohort studies or clinical trials it serves as a robust and practical measure that gives insights into the HIV reservoir that is different from the measure of residual VLs.

Reply: We thank the reviewer for this very positive assessment of our work.

Specific comments:

1. It is particularly interesting that non-white ethnicity associated with lower HIV-1 reservoir

size at 1.5 years after ART initiation. It would be helpful to have a breakdown of the non-white group of individuals – from where in Africa/elsewhere do they come from? As host genetic variability is most extensive in Africa - it would be important that this is at least documented, as may be important for future studies.

Reply: We agree with the reviewer that a breakdown of regions of origin within the non-white study participants is of interest for future studies. We analyzed the regions of origin of white and non-white study participants and included this analysis as new **Supplementary Fig. S1** (please see below). While a large group of non-white individuals had nationalities from South-Eastern Asia (51, 23.5%) there were 103 individuals with nationalities from different parts of Africa: 40 from Middle Africa (18.4%), 40 from Eastern Africa (18.4%) and 23 from Western Africa (10.6%). This reiterates that our study population represents a real-life, i.e., very diverse population. We indicate this new analysis in the results section in line 117: “..., and individuals of non-white ethnicity (n = 217, for a breakdown of regions of origin see **Supplementary Fig. S1**).”

Figure S1. Regions of origin of study participants. 724 (86.2%) individuals with white ethnicity were from Western Europe. The largest group of non-white individuals were from South-Eastern Asia (n = 51, 23.5%), followed by Middle Africa (n = 40, 18.4%), Eastern Africa (n = 40, 18.4%), South America (n = 28, 12.9%), and Western Africa (n = 23, 10.6%).

2. That non-B subtypes had lower HIV-1 DNA levels compared to HIV-1 subtype B in the white ethnic group – suggests viral subtype differences may be an important determinant in HIV-1 reservoir size. This needs some discussion, are there data from studies that have directly looked at disease progression with different HIV-1 subtypes in the same population that could support this finding.

Reply: Indeed, differences in the HIV-1 reservoir size might be partly dependent on the HIV-1 subtype. We did not want to put too much emphasis on this finding, since in the multivariable analysis we only observed a non-significant trend of HIV-1 non-B subtype towards having a small reservoir (**Fig. 3a**). Nevertheless, we further explored this topic and found differences of HIV-1 reservoir size between HIV-1 B and non-B subtypes in individuals of white ethnicity ($p = 0.017$, **Supplementary Fig. S7b**). However, beware: This analysis was not adjusted for covariables and therefore, should be interpreted with caution.

Based on the reviewer's suggestion, we further investigated this by performing the regression analysis presented in **Fig. 3** including only individuals of white ethnicity with available HIV-1 subtype information (including correction for all other covariables). We observe a trend towards a lower HIV-1 reservoir, however, it was not significant in the multivariate analysis ($p=0.059$, **new Supplementary Fig. S9**) (please see below). This indicates, that the effect observed in the unadjusted analysis in **Fig. S7b** might be partially mediated by the effect of HIV-1 subtype on other covariables, for example viral load.

- To add this additional analysis, we adjusted the sentence in the results section, lines 199-202, as follows:
“Furthermore, the effect of HIV-1 non-B subtypes was **still present** when we restricted our analysis to individuals with white ethnicity ($p = 0.017$, **Supplementary Fig. S7b**) and even in a multivariable analysis a trend of a lower HIV-1 reservoir size in individuals infected with HIV-1 non-B subtypes persisted ($p = 0.059$, **Supplementary Fig. S9**).”
- We also now discuss these findings in lines 353-357:
“... reported lower viral set-points in people from African origin. **Although not significant, we observed trends of lower HIV-1 reservoir size and faster HIV-1 reservoir decay in individuals infected with HIV-1 non-B subtypes that were not driven by ethnicity. These findings accentuate the need to further explore viral subtype- and ethnicity-dependent pathogenetic mechanisms (Bhargava et al., 2014; Taylor et al., 2008).**”

Figure S9: Analysis of determinants of HIV-1 reservoir size and long-term dynamics as shown in Fig. 3 restricted to individuals of white ethnicity and with available HIV-1 subtype information. 581 individuals were infected with HIV-1 subtype B and 133 individuals with HIV-1 non-B subtypes. (a) Coefficient plot showing covariables associated with total HIV-1 DNA levels 1.5 years after initiation of ART. Viral load <50 HIV-1 RNA copies/ml plasma or low-level viremia refer to the time from 180 days after initiation of ART to the first HIV-1 DNA quantification. Reference was defined as initiation after first year of HIV-1 infection, transmission group MSM, white ethnicity and HIV-1 subtype B. (b) Coefficient plot showing covariables associated with the decay of total HIV-1 DNA levels. Corrected for initial HIV-1 DNA levels using a spline. Viral load <50 HIV-1 RNA copies/ml plasma or low-level viremia or blips refer to the time between the first and third sample, i.e. 1.5-5.4 years after initiation of ART. Baseline as in panel a. ART, antiretroviral therapy; MSM, men who have sex with men; HET, heterosexual; PWID, people who inject drugs; transmission group other includes unknown and transfusion; time to viral suppression refers to time taken for viral load to drop below 50 HIV-1 RNA copies/ml plasma; CD4+ cell count was measured per 200 cells/ μ l blood.

3. Did the authors directly test whether there are differences in HIV-1 reservoir measures between men and women? It has been shown that women have better control of HIV-1 in early untreated infection (lower VLs) but in chronic infection progress more quickly compared to men – so in the long term progress to AIDS in similar time. This would suggest that perhaps women might have an additional advantage in achieving a low HIV-1 reservoir if treated early compared to men.

Reply: We thank the reviewer for raising this point. In our initial analyses we performed a regression analysis including sex and transmission group as different variables (please see figure below). Women had a significantly lower reservoir size 1.5 years after initiation of ART in the univariable analysis. However, this effect disappeared in the multivariable analysis indicating that it was driven by possible confounding factors. We noted that sex and transmission group were highly correlated in our study population (due to the large fraction of MSM) and thus, we combined sex and transmission group for our main analyses.

Sex as a determinant of HIV-1 reservoir size and long-term dynamics. (a) Coefficient plot showing covariables associated with total HIV-1 DNA levels 1.5 years after initiation of ART. Viral load <50 HIV-1 RNA copies/ml plasma or low-level viremia refer to the time from 180 days after initiation of ART to the first HIV-1 DNA quantification. Reference was defined as viral load, plasma HIV-1 RNA below 50 copies/ml, initiation after first year of HIV-1 infection, transmission group MSM, white ethnicity and HIV-1 subtype B. **(b)** Coefficient plot showing covariables associated with the decay of total HIV-1 DNA levels. Corrected for initial HIV-1 DNA levels using a spline. Viral load <50 HIV-1 RNA copies/ml plasma or low-level viremia or blips refer to the time between the first and third sample, i.e. 1.5-5.4 years after initiation of ART. Baseline as in panel a. ART, antiretroviral therapy; MSM, men who have sex with men; HET, heterosexual; PWID, people who inject drugs; transmission group other includes unknown and transfusion; time to viral suppression refers to time taken for viral load to drop below 50 HIV-1 RNA copies/ml plasma; CD4+ cell count was measured per 200 cells/ μ l blood.

4. The smaller reservoir size in the PWID transmission group after correction for IFN treatment for HCV – how can one explain this finding? Not sure if there are any data on other modes of transmission e.g. blood transfusion – where, as with PWIDs, one would expect infection occurs with a quasispecies of viral variants. This is very different to mucosal transmission - where a bottleneck occurs with transfer of one or very few viral variants with other modes of transmission (sexual, mother-to-child transmission).

Reply: This is another interesting point raised by the reviewer. For 25 of the 46 individuals with other modes of transmission, the mode of transmission remains unknown or was not disclosed. Only 13 individuals were presumably infected via blood products or clotting factors against hemophilia. 8 individuals reported other sources. However, we hypothesized that our findings may reflect a survival bias. To investigate this hypothesis, we examined the years of recruitment to the SHCS per transmission group. Interestingly, a substantial number of individuals within the PWID group was HIV-1 infected before potent combination ART was available, i.e., we have probably selected for long-term non-progressors among the PWID group, who are known to have lower HIV-1 reservoir sizes (Blankson et al., 2007) (new **Supplementary Fig. S11**). This is now described in lines 215-220:

“Of note, a substantial fraction of the PWID group was HIV-1 infected years before potent ART became available (**Supplementary Fig. S11**). Thus, the association PWID with smaller HIV-1 reservoir size 1.5 years after initiation of ART might reflect a survival bias, i.e., an overrepresentation of long-term non-progressors among the PWID group, known to have smaller HIV-1 reservoir sizes (Blankson et al., 2007).”

Along this new analysis, we discovered that none of the individuals were perinatally infected with HIV-1, thus, we deleted «perinatal transmission» in all figure legends where this was listed as potential other mode of transmission.

Figure S11. Barplot showing the transmission groups per year of SHCS registration. Width of the bars correspond to the number of registrations per year on a log scale. MSM,

men who have sex with men; HET, heterosexual; PWID, people who inject drugs; transmission group other includes unknown and transfusion.

5. Another intriguing finding is the subset of individuals (26.6%) who did not achieve a negative slope of their total HIV-1 DNA levels after ART initiation. This finding warrants more discussion – what were the characteristics of this specific group, does anything stand out, what could explain this outcome for a substantial proportion of the cohort?

Reply: We thank the reviewer for this excellent suggestion, which helped to strengthen the evidence presented in our manuscript. To explicitly define characteristics of the group of individuals with a positive slope, we performed a multivariable logistic regression using the classification into positive / negative slope as outcome variable. This analysis has the advantage of providing odds ratios, which represent the constant effect of covariables on the likelihood that a positive slope occurred. However, at the same time, this analysis had the disadvantage of loss of statistical power (compared to our main analysis) because binarizing the slopes implies that a part of the information contained in these values is not taken into account (i.e., how strongly positive or negative a slope is). Along with a trend observed in our main analysis (**Fig. 3b**), we found that non-white ethnicity was associated with having a positive slope (OR = 1.657, $p = 0.036$). Thus, being an individual of non-white ethnicity was associated with having low HIV-1 DNA values 1.5 years after the initiation of ART and at the same time having a positive HIV-1 DNA slope (even after adjustment for HIV-1 DNA 1.5 years after the initiation of ART). In absolute numbers, 84/217 (38.7%) individuals of non-white ethnicity had a positive slope. Also in line with our main analysis, viral blips were a distinct characteristic of the individuals with a positive slope (OR = 1.618, $p = 0.0185$). In general, one could hypothesize that in this group of individuals with increasing slope, clonal expansion occurred more frequently than in the group with a decreasing slope, or that residual replication was responsible for increased total HIV-1 DNA levels.

Further, the reviewers comment helped addressing the question of reviewer 1, asking whether assay and sample variability could drive the high fraction of positive slopes (please find our reply to comment 4 of reviewer 1 above), because the logistic regression analysis clearly shows that having a positive slope is not due to assay variability only (if this were the case we would not expect to observe the associations shown in **Fig. S12** below).

We have extended the sentence in line 252 in our results section, to indicate the interesting association of non-white ethnicity with a positive slope shown in **Fig. S12**:

“Notably, non-white ethnicity showed a trend towards slower decay of the HIV-1 reservoir ($p = 0.075$, Fig. 3b) and was at the same time significantly associated with having a positive slope in a logistic regression model (OR = 1.657, $p = 0.036$, **Supplementary Fig. S12**).”

Figure S12: Odds ratio for positive HIV-1 reservoir decay slope. Corrected for initial HIV-1 DNA levels using a spline. Viral load <50 HIV-1 RNA copies/ml plasma or low-level viremia or blips refer to the time between the first and third sample, i.e. 1.5-5.4 years after initiation of ART. Baseline as in panel a. ART, antiretroviral therapy; MSM, men who have sex with men; HET, heterosexual; PWID, people who inject drugs; transmission group other includes unknown and transfusion; time to viral suppression refers to time taken for viral load to drop below 50 HIV-1 RNA copies/ml plasma; CD4+ cell count was measured per 200 cells/ μ l blood.

Reviewer #3:

General

The authors have explored associations of the HIV-1 reservoir size and long-term decay with various host and viral factors in a large scale observational study. This complex interplay is shown in an in-depth analysis in an impressive patient cohort ($n=1057$) and followed for an extensive time with HIV-1 DNA measured at 3 timepoints: 1.5 years, 3.5 years and 5.4 years post-ART initiation ($n=645$) and in some patients ($n=412$) 4th timepoint was available at 10 years after ART was initiated. This study provides a comprehensive analysis of determinants of the HIV-1 reservoir size and dynamics in patients on suppressive ART. It is unique in such a large scale. However, no new conclusions are drawn in this study, neither none of the proposed biological mechanisms explaining interplay between the residual viremia and HIV-1 DNA reservoir are being explored. Thus, this is purely a descriptive study with the most

prominent strength of a well characterized cohort of an impressive number of individuals included.

Most important findings of the study are the shown associations between viral blips and low-level viremia with slower reservoir decay, timing of ART initiation, pre-ART viral load. Longer time on ART was shown to be a predictor for a smaller reservoir. Furthermore, significant decrease in the size of HIV-1 DNA reservoir was shown 1.5 to 5.4 years post-ART, which plateaus afterwards. And interestingly, non-white ethnicity seemed to be associated with a smaller reservoir size.

Reply: We thank the reviewer for his/her appreciation of the concept of our study and the comprehensiveness and uniqueness of the underlying data-set. Regarding the concerns raised, please see our detailed responses below.

Major comments

Lack of novelty, purely observational study. The size of the analyzed patient group and the number of samples analyzed per patient is impressive and a study of this magnitude is able to draw conclusions with a high statistical power.

Reply: We must disagree with the reviewer on both, the lack of novelty of our study and the statement that our study was purely descriptive (made in the general remark). To specifically address the reviewer's first concern, the following elements illustrate why we are confident that our study provides novel and important data, and reveals biologically relevant and novel observations:

- This was the first study to examine the HIV-1 reservoir size and its long-term dynamics over extensive follow-up periods in more than 1,000 individuals.
- This was the first study to examine the HIV-1 reservoir size and its long-term dynamics in a population-based cohort representative of a real-world population.
- The small study sizes of previously analyzed populations did not allow to conduct analyses with sufficient statistical power to include and adjust for a variety of relevant variables (and hence to show the independent effect of these variables). For the current study, the extensive data of the underlying SHCS did allow us to correct for many known and suspected confounding factors.
- This was the first study showing a significant association of viral blips and HIV-1 reservoir size and decay. Our observational study design allowed us to assess the relationships of residual viremia (low level viremia) or viral blips with HIV-1 reservoir size and long-term dynamics under successful ART. We could show that viral blips as well as low-level viremia, slowed down the HIV-1 reservoir decay.

The reviewer is correct, that this was an observational study with a large sample size and accordingly large statistical power. While we agree that our study was observational, it is not correct that it was purely descriptive (as the reviewer suggests in the general remark). With our regression models we could clearly identify several key determinants of both HIV-1 reservoir size and decay. Such associations found from observational data are an extremely valuable source of evidence and indications for causal relationships. While it is true that in observational studies such inferences are always subject to potential confounding, it is one strength of our study that this analysis was conducted in the context of the extremely data-rich

SHCS allowing to adjust for the crucial confounding factors. One additional strength of this research design is that it captures the real-life situation and is representative of the very heterogeneous population of people living with HIV. It should be noted that in accordance with these strengths, effects found in observational (cohort) studies are considered as the second highest quality of evidence (after randomized controlled trials), when it comes to clinical decisions and guidelines (Saag et al., 2018), while evidence from experimental studies ranges below in the hierarchy of evidence. Thus, we do not think that in terms of quality of evidence our study is purely descriptive in the sense of not informing about relevant effects. Just to take an example, the strong, robust, and novel evidence for effect of viral blips will be relevant for patient selection in the context of future proof-of-concept cure studies. In addition, it will trigger additional experimental/mechanistic studies to further investigate causality. In theory, if no differentiation between patients with and without viral blips is made, results of such cure studies could be falsely negative because of confounding by residual replication, if viral blips are in part caused by residual replication.

Lines 216-219: Showing smaller reservoir size associated with HIV-1 non-B subtype, where indeed evaluation of the PCR assay needs to be performed. How were the HIV-1 full-length sequences obtained? This would need to be described better in the manuscript. Here, an in silico analysis is mentioned, where no PCR mismatches were detected in 568/1368 individuals. This needs a better elaboration. Which patients are representing these 568 patients?

Reply: We agree with the reviewer that this merits further explanation. From 568 of the initially included 1.382 (sorry, 1368 is a typo) individuals, plasma samples were available in the SHCS biobank prior to initiation of ART. Near full-length genome sequencing was performed HIV-1 using those plasma samples. Considering the 1,057 individuals finally included in the analyses, we obtained near HIV-1 full-length genome sequenced from 512.

- Thus, we corrected these figures in line 204:
... for 512/1,057 individuals ...”
- Furthermore, we added the following paragraph in the material and method section in line 460:

“HIV-1 near full-length genome sequencing

“Plasma samples from the last lab visit before the initiation of ART were available for 512/1,057 individuals. If it was previously known from Sanger sequencing for genotypic resistance testing that the individual was infected with HIV-1 subtype B, sequencing was done according to the protocol published by Di Giallonardo and colleagues (Giallonardo et al., 2014), while in the case of non-B or unknown HIV-1 subtype sequencing was performed according to the protocol published by Gall and colleagues (Gall et al., 2012).”

In Figure 4, authors are illustrating the complex interplay of residual viremia and HIV-1 reservoir size and its long-term dynamics, with two possible scenarios explaining this interplay and long-term dynamics. However, authors do not bring here an analysis studying these proposed mechanisms. The study remains purely descriptive. Adding a more mechanistic exploration of the HIV-1 reservoir would enhance this study substantially. Would that be possible with the samples collected?

Reply: While we must disagree with the reviewer that our study remained purely descriptive (as discussed in our reply to the reviewers' first comment), we agree with the reviewer, that it would be interesting to explore the biological mechanism behind the observed patterns in detail. Further, this point comes to the key aim of our study, which is to guide relevant future directions for HIV-1 reservoir and cure research. We were able, with a systematic analysis of a large population-based cohort, to detect and quantify patterns of interactions of the HIV-1 reservoir with residual viremia – for this purpose, our study design was considerably different from a study designed to investigate a specific biological mechanism in detail. Further, even though, we could not fully disentangle causality between residual viremia and the HIV-1 reservoir, we can gain a considerable understanding of their interaction from our study. Our longitudinal quantification of the HIV-1 reservoir enabled us to separate residual viremia and HIV-1 reservoir size/decay by time and thus, we can to some extent separate cause and effect, as we aimed to clarify in **Fig. 4c**.

Of course, future studies with the focus on mechanistic explorations of viral blips, low-level viremia, and the proposed driving factors for halting the decay of the latent reservoir are of very great interest. Thus, further studies are planned for a subsample of the 1,057 individuals. We have already planned proviral full-length genome sequencing to search for traces of evolution and clonal expansion within a smaller cohort. Furthermore, also HIV-1 full-length genome sequencing of viral blips would be of great interest, to investigate their origins. However, these projects are clearly outside the scope of the current study.

In 1057 of 1166 patients, total HIV-1 DNA quantification was achieved. What was the cause why it did not work in the other samples? Low quality of samples? Or were these all samples with a low HIV-1 DNA level (not quantifiable), so are patients excluded with a very low HIV-1 DNA reservoir? (biased towards patients that started ART very early?)

Reply: We acknowledge the reviewers' concern and the need for clarification. Importantly, for all individuals with unsuccessful quantification of total HIV-1 DNA, technical problems occurred with one to all PBMC samples: Either, DNA isolation or DNA sonication failed, and thus, there was no DNA in the subsequent ddPCR (i.e., no signal for CCR5 nor HIV-1), or the ddPCR run failed due to, e.g., no droplet formation or no droplet counting. Since we used almost the entire isolated DNA for ddPCR to achieve a high sensitivity, we were not able to repeat any of those samples. Taken together, the reasons for failures to measure viral DNA were unrelated to the HIV-1 DNA level and hence a bias towards patients with a very low HIV-1 reservoir is highly unlikely.

To clarify this, we added the following sentence in the results section, lines 111-114:

“Since technical problems, for instance, failed DNA isolation, were the reason for all unsuccessful quantifications of total HIV-1 DNA, failures to measure total HIV-1 DNA are very unlikely to have introduced a bias to our study population of 1,057 individuals.”

Is there more information available about the Fiebig stages of the patients that initiated ART within 1 year? This could give interesting additional data.

Reply: We thank the reviewer for raising this interesting point. For 16 patients, who are enrolled in both, the SHCS and the Zurich Primary HIV Infection Cohort Study (ZPHI: a longitudinal study enrolling patients with known infection dates during primary HIV infection (Braun et al., 2015; Rieder et al., 2011)), Fiebig stages at the day of initiation of ART were available. An analysis of these patients is now included in the result section in line 180:

“For a subset of 16 individuals with known Fiebig stages (Fiebig et al., 2003) at the day of initiation of ART, the median HIV-1 reservoir was strikingly low for two individuals in Fiebig stage II and lower for individuals in Fiebig stages IV-VI as compared to the population level median (**Supplementary Fig. S5**).”

Figure S5. Beanplot of total HIV-1 DNA levels 1.5 years after initiation of ART stratified by Fiebig stage. Classification of Fiebig stage was possible in 17/173 individuals starting ART within the first year of HIV-1 infection. The individual observations are shown as small black lines in a one-dimensional scatter plot. Overlaid is the estimated density of the distributions (filled in pink) and the median is depicted by a horizontal black line. The gray line indicates the population level median total HIV-1 DNA 1.5 years after initiation of ART.

Minor comments

An interesting finding mentioned is the significant effect of non-white ethnicity, however, it is not clearly mentioned whether possible confounding effect may be biasing this finding.

Reply: We acknowledge the reviewers’ concern. Indeed, the significant association of non-white ethnicity with a small HIV-1 reservoir size 1.5 years after the initiation of ART is very interesting. From the pattern observed in the regression analysis presented in Fig. 3a, i.e., similar effect sizes for the univariable and multivariable analysis, we can infer that there was no (strong) confounding by the covariables included in the regression analysis. From this we can conclude, that there was no confounding by suspected confounders like HIV-1 subtype, CD4+ cell counts, log₁₀ HIV-1 RNA pre-ART, or initiation of ART within/after first year of HIV-1 infection. However, as with any observational study, associations like these must be interpreted cautiously, because of the potential for unmeasured confounding factors (as already stated in our discussion). Also related to this comment are our answers to comments 1 and 2 of reviewer 2 (please see above).

Figure 1b: The figure can be misleading: it insinuates that there is a blip around every HIV-1 DNA quantification.

Reply: We thank the reviewer for this remark and agree that this might have been misleading. We removed the blips from **Fig. 1b** (please see below).

What is the range of copies/ml of the blips? (Maximum level?)

Reply: Blips were solely defined by the presence of a measurements ≥ 50 HIV-1 RNA copies/ml plasma, which was preceded and followed by measurements < 50 HIV-1 RNA copies/ml plasma. Any subsequent viral load measurement ≥ 50 HIV-1 RNA copies/ml plasma within 30 days of a viral blip was considered to be part of the same viral blip (Young et al., 2015). We did not impose a numerical upper limit for the blip height.

Considering multiple blips per individual, we found a median blip height of 77 (IQR: 58-128) RNA copies/ml plasma before the first and a median blip height of 83 (IQR: 61-150 HIV-1) RNA copies/ml plasma between the first and third total HIV-1 DNA measurement. 7 patients had inexplicable viral blips of $> 20,000$ HIV-1 RNA copies/ml plasma (2 before the first and 5 between the first and third total HIV-1 DNA measurement). We verified that these patients did not interrupt ART or experienced virologic failure by checking the clinical charts. We did not exclude those individuals, since we could not find an explanation fulfilling an exclusion criterium for these high values. One explanation might be that in a large cohort of $> 1,000$ individuals longitudinally followed-up for several years in multiple clinical centers the possibility of mislabeling or intermingling some rare samples/clinical data cannot be avoided entirely. However, we repeated our analyses excluding those 7 patients, which did not change the results (please see the figure below, which we could add to our supplementary material if the reviewer/editor thinks this is helpful). Along these lines, please note, that since in our regression analysis the presence of viral blips or low-level viremia was included as a categorical variable, viral blip height did not drive the observed pattern.

Size of viral blips as a determinant of HIV-1 reservoir size and long-term dynamics – excluding 7 patients with high viral blips. (a) Coefficient plot showing covariables associated with total HIV-1 DNA levels 1.5 years after initiation of ART. Viral load <50 HIV-1 RNA copies/ml plasma or low-level viremia refer to the time from 180 days after initiation of ART to the first HIV-1 DNA quantification. Reference was defined as no viral blips, initiation after first year of HIV-1 infection, transmission group MSM, white ethnicity and HIV-1 subtype B. (b) Coefficient plot showing covariables associated with the decay of total HIV-1 DNA levels. Corrected for initial HIV-1 DNA levels using a spline. Viral load <50 HIV-1 RNA copies/ml plasma or low-level viremia or blips refer to the time between the first and third sample, i.e. 1.5-5.4 years after initiation of ART. Baseline as in panel a. ART, antiretroviral therapy; MSM, men who have sex with men; HET, heterosexual; PWID, people who inject drugs; transmission group other includes unknown and transfusion; time to viral suppression refers to time taken for viral load to drop below 50 HIV-1 RNA copies/ml plasma; CD4+ cell count was measured per 200 cells/ μ l blood.

PWID: abbreviation is not explained in the text, only in the figure.

Reply: We corrected that in line 207:
“... people who inject drugs (PWID) ...”

Figure 4c: This figure is referred to only in lines 299-300. The figure could use a more detailed explanation in the text and make the link more comprehensible with the text.

Reply: We acknowledge the reviewers' concern and agree that previously the explanations for **Fig. 4c** were insufficient. This is in line with a comment by reviewer 1. We improved the comprehensibility of **Fig. 4c** within the figure, in the figure legend, and by additional explanations in the text. For details please see our answer to other concern #5 of reviewer 1.

The clinical relevance of 1.93 and 1.79-fold difference at slope values of -0.06 -> -0.04 should be more elaborated.

Reply: The clinical relevance of those specific differences cannot be answered yet and needs further investigations. Here, we describe the different HIV-1 reservoir decay slopes in different groups of individuals, however, we find it too early to draw conclusions on potential clinical relevance. Apart from the exact values, the range of numbers illustrates a strong effect of viral blips and low-level viremia in a very well-treated cohort. And this highlights that viral blips may have an impact in the context of cure studies. Clearly, more research is needed in the future to identify the exact biological nature of blips and its impact on clinical medicine. If requested, we could delete the numbers 1.79- and 1.93-fold in line 269.

Additional clarifications:

- During the revisions, we added several new supplementary figures and adjusted the numbering in the manuscript and the supplementary material accordingly.
- We made the following clarification about the definition of blips and low-level viremia in our Methods section, line 497: "If an individual exhibited both, viral blips and low-level viremia, we classified this as low-level viremia for our analysis."
- We clarified that we display 95% confidence intervals in coefficient plots by adding "and 95% confidence intervals" in the figure legend of **Fig. 3**, lines 849 and 854, and all coefficient plots in our supplementary material.
- We found that the sentence in line 279 was misleading. The sentence reads now: "Interestingly, the frequency of viral blips was decreasing with time after initiation of ART and was generally lower in individuals initiating ART within the first year of HIV-1 infection-~~smaller HIV-1 reservoir size and faster decay~~ (**Supplementary Fig. S1744**)."
- While revising the manuscript, we noted that we have not explained, how we dealt with missing values. The regression analysis presented is a complete case analysis, while a sensitivity analysis using multiple imputation for missing values was added to the supplementary material. We clarified this in our methods section in lines 476-486: "The regression analysis represents a complete case analysis. As a sensitivity analysis, we performed an equivalent regression analysis utilizing multiple imputation for missing values (98 log₁₀ HIV-1 plasma RNA pre-ART values, 34 CD4+ cell count pre-ART values, and 14 individuals had no HIV-1 RNA measurement between 180 days after initiation of ART and the time point of the 1st total HIV-1 DNA measurement) using the R package mice (van Buuren and Groothuis-Oudshoorn, 2011). A sensitivity analysis can be found in the supplementary material (**Supplementary Fig. S18**). Results of the multivariable analysis remained qualitatively unchanged, apart from (i) the association of non-white ethnicity with a slow HIV-1 reservoir decay, for which the

p-value changed from 0.075 to 0.034 and (ii) the association of female PWID with the decay of the HIV-1 reservoir, which lost significance.”

Figure S18: Size of viral blips as a determinant of HIV-1 reservoir size and long-term dynamics – missing values were replaced using 10 multiple imputations generated by the R package MICE, which generates multivariate imputations by chained equations. (a) Coefficient plot showing covariables associated with total HIV-1 DNA levels 1.5 years after initiation of ART. Viral load <50 HIV-1 RNA copies/ml plasma or low-level viremia refer to the time from 180 days after initiation of ART to the first HIV-1 DNA quantification. Reference was defined as no viral blips, initiation after first year of HIV-1 infection, transmission group MSM, white ethnicity and HIV-1 subtype B. (b) Coefficient plot showing covariables associated with the decay of total HIV-1 DNA levels. Corrected for initial HIV-1 DNA levels using a spline. Viral load <50 HIV-1 RNA copies/ml plasma or low-level viremia or blips refer to the time between the first and third sample, i.e. 1.5-5.4 years after initiation of ART. Baseline as in panel a. ART, antiretroviral therapy; MSM, men who have sex with men; HET, heterosexual; PWID, people who inject drugs; transmission group other includes unknown and transfusion; time to viral suppression refers to time taken for viral load to drop below 50 HIV-1 RNA copies/ml plasma; CD4+ cell count was measured per 200 cells/ μ l blood.

REVIEWERS' COMMENTS:

Reviewer #1 (Remarks to the Author):

The revised manuscript addressed most of my concerns. The authors provided a careful reply to the reviewer's comments. This is a strong manuscript.

Remaining concerns:

1. Discussion, lines 357-359: "Notably, total HIV-1 DNA levels did not decrease after initiation of ART in 281/1,057 (26.6%) individuals over a median of 5.4 years of suppressive ART."

This looks like a typo. The results show "did not decrease beyond 1.5 years after initiation of ART in ...".

2. Discussion, line 312-319: "In our study, lack of adherence can be ruled out as a major factor for viral blips since virologic failure and treatment interruptions were exclusion criteria and self-reported adherence was high in the study population, i.e., 920/1,051 (87%) individuals never reported to have missed more than 1 pill in the previous month..."

"Lack of adherence can be ruled out as a major factor for viral blips" is a very strong statement, and the authors provide insufficient support for this statement. 1) In their dataset, 130 (12%) of participants experience viral blips. This is below the 13% number of participants who reported non-adherence.

2) The fact that self-reported adherence is a predictor for lower risk of clinical events just means that there is some correlation between self-report and true adherence; it does not prove that lack of adherence can be ruled out as a major factor for viral blips. To "rule out", investigators would have to have a VERY reliable measure of adherence, and then show that most blips are not preceded by non-adherence. This data is simply not available. I still suggest that authors tone down their statement.

The main argument in favor of the author's statement is that "upon adjusting with the proxy of adherence, the effect-sizes for blips and low-level viremia changed only marginally". This does suggest that self-reported adherence is not a strong predictor of viral blips, but the evidence does not rise to the level of "lack of adherence can be ruled out as major factor".

Reviewer #2 (Remarks to the Author):

The authors have very adequately addressed reviewers comments and made relevant changes to the manuscript, all which have added to further strengthen the manuscript. As highlighted previously, few studies have directly asked the question of HIV-1 DNA decay during ART and how this might relate to residual viremia; those that have were conducted on substantially smaller numbers of patients compared to the current study. This is a very substantial body of work that will make an important contribution to the field.

Specific comments:

An additional point given the breakdown that was now provided of the non-white individuals in the study. Within this group one would expect individuals of Asian and African origin to possibly be quite different (genetics, environment). Since the last review of this manuscript the following paper has been published: Gossez et al, AIDS 2019: Virological remission after ART interruption in female African seroconverters.

This particular paper from the SPARTAC Trial investigators studied 82 patients from South Africa and Uganda, all were female and all were mainly infected with non-B (most were C) subtypes. They found that Africans had a longer period of viral control before rebound compared to non-Africans *HR 3.90, 1.75-8.71, $p < 0.001$). The possible contribution of gender and subtype could not be accounted for in this study as the African patients were all female - and they were compared to non-African (subtype B participants from the UK, Italy, Ireland, Australia and Brazil). Nonetheless, the findings are interesting

in light of the lower HIV-1 DNA levels in your non-white group (and lower DNA levels have been associated with longer time to rebound if ART is stopped). Would it be possible to look within the non-white group and see if the Africans are driving this association (not sure numbers will be sufficient but think worth a look, and also look at the gender distribution). This could provide very important insights into population- or even sex-based differences. I acknowledge the previous response to the question raised on gender, but here there might be a skewing of higher representation of women in a particular population group within the non-white category.

The same would apply to the 26.6% who did not achieve a negative slope of total HIV-1 DNA levels – as was mentioned these were found to be of non-white ethnicity. Are they perhaps more likely African, female?

Reviewer #3 (Remarks to the Author):

The authors have addressed systematically and precisely each point raised, strongly supporting the uniqueness and high importance of their large-scale observational study. Indeed, as stated by the authors, a study of this scale ($n > 1000$) in HIV-1 reservoir size and dynamics, with the strength in the exceptionally rich dataset allowing to correct for many confounding factors, has not been performed before and provides valuable insights for future HIV-1 reservoir and cure studies design as well as for the following in-depth mechanistic studies, which have been already planned by the authors.

Furthermore, we acknowledge the authors for the adjustments made in the manuscript, including the added paragraph on HIV-1 near full-length genome sequencing and the figure of HIV-1 reservoir in patients stratified by Fiebig stage at initiation of ART.

Point-by-point reply round 2

Our point by point reply contains **reviewer's comments in blue**, answers in black, and **changes in the manuscript in brown**.

Reviewer #1:

The revised manuscript addressed most of my concerns. The authors provided a careful reply to the reviewer's comments. This is a strong manuscript.

Reply: We are grateful for the reviewers' favorable assessment of our revised manuscript.

Remaining concerns:

1. Discussion, lines 357-359: "Notably, total HIV-1 DNA levels did not decrease after initiation of ART in 281/1,057 (26.6%) individuals over a median of 5.4 years of suppressive ART." This looks like a typo. The results show "did not decrease beyond 1.5 years after initiation of ART in ...".

Reply: We appreciate the reviewers' observation and adjusted the sentence in lines 412-414 accordingly:

"Notably, total HIV-1 DNA levels did not decrease **after initiation of ART** in 281/1,057 (26.6%) **during the years 1.5 to 5.4 after initiation of suppressive ART** **individuals over a median of 5.4 years of suppressive ART.**"

2. Discussion, line 312-319: "In our study, lack of adherence can be ruled out as a major factor for viral blips since virologic failure and treatment interruptions were exclusion criteria and self-reported adherence was high in the study population, i.e., 920/1,051 (87%) individuals never reported to have missed more than 1 pill in the previous month..."

"Lack of adherence can be ruled out as a major factor for viral blips" is a very strong statement, and the authors provide insufficient support for this statement. 1) In their dataset, 130 (12%) of participants experience viral blips. This is below the 13% number of participants who reported non-adherence.

2) The fact that self-reported adherence is a predictor for lower risk of clinical events just means that there is some correlation between self-report and true adherence; it does not prove that lack of adherence can be ruled out as a major factor for viral blips. To "rule out", investigators would have to have a VERY reliable measure of adherence, and then show that most blips are not preceded by non-adherence. This data is simply not available. I still suggest that authors tone down their statement.

The main argument in favor of the author's statement is that "upon adjusting with the proxy of adherence, the effect-sizes for blips and low-level viremia changed only marginally". This does suggest that self-reported adherence is not a strong predictor of viral blips, but the evidence does not rise to the level of "lack of adherence can be ruled out as major factor".

Reply: We acknowledge the reviewers' concern and toned down the statement under consideration. In the discussion, line 363-365 read now:

“In our study, lack of adherence is unlikely to be a major driver of ~~can be ruled out as a major factor for~~ viral blips since virologic failure and treatment interruptions were exclusion criteria and self-reported adherence was high in the study population, ...”

Reviewer #2:

The authors have very adequately addressed reviewers comments and made relevant changes to the manuscript, all which have added to further strengthen the manuscript. As highlighted previously, few studies have directly asked the question of HIV-1 DNA decay during ART and how this might relate to residual viremia; those that have were conducted on substantially smaller numbers of patients compared to the current study. This is a very substantial body of work that will make an important contribution to the field.

Reply: We would like to thank the reviewer for the positive evaluation of our revised manuscript.

Specific comments:

An additional point given the breakdown that was now provided of the non-white individuals in the study. Within this group one would expect individuals of Asian and African origin to possibly be quite different (genetics, environment). Since the last review of this manuscript the following paper has been published: Gossez et al, AIDS 2019: Virological remission after ART interruption in female African seroconverters.

*This particular paper from the SPARTAC Trial investigators studied 82 patients from South Africa and Uganda, all were female and all were mainly infected with non-B (most were C) subtypes. They found that Africans had a longer period of viral control before rebound compared to non-Africans *HR 3.90, 1.75-8.71, p<0.001). The possible contribution of gender and subtype could not be accounted for in this study as the African patients were all female - and they were compared to non-African (subtype B participants from the UK, Italy, Ireland, Australia and Brazil). Nonetheless, the findings are interesting in light of the lower HIV-1 DNA levels in your non-white group (and lower DNA levels have been associated with longer time to rebound if ART is stopped). Would it be possible to look within the non-white group and see if the Africans are driving this association (not sure numbers will be sufficient but think worth a look, and also look at the gender distribution). This could provide very important insights into population- or even sex-based differences. I acknowledge the previous response to the question raised on gender, but here there might be a skewing of higher representation of women in a particular population group within the non-white category.*

The same would apply to the 26.6% who did not achieve a negative slope of total HIV-1 DNA levels – as was mentioned these were found to be of non-white ethnicity. Are they perhaps more likely African, female?

Reply: We thank the reviewer for pointing out this interesting new publication. Indeed, the findings of Gossez et al.¹ that the period of viral control was longer in Africans compared to non-Africans corresponds well with the lower HIV-1 reservoir 1.5 years after initiation of ART found in non-white participants of our study. We have already planned in-depth analyses on the contribution of human genomics to the size and decay of the HIV-1 reservoir and thus, prefer not to include more analyses on that topic to our manuscript. However, we now discuss the new publication mentioned by the reviewer in line 403-408 of our manuscript:

“An interesting finding of our study was the independent and significant effect of non-white ethnicity on the size of total HIV-1 DNA levels 1.5 years after initiation of ART. This result is consistent with Müller et al.² who reported lower viral set-points in people from African origin

and Gossez et al.¹ who found a higher probability of virological remission following treatment interruption among female Africans compared to female non-Africans.”

Reviewer #3:

The authors have addressed systematically and precisely each point raised, strongly supporting the uniqueness and high importance of their large-scale observational study. Indeed, as stated by the authors, a study of this scale (n>1000) in HIV-1 reservoir size and dynamics, with the strength in the exceptionally rich dataset allowing to correct for many confounding factors, has not been performed before and provides valuable insights for future HIV-1 reservoir and cure studies design as well as for the following in-depth mechanistic studies, which have been already planned by the authors.

Furthermore, we acknowledge the authors for the adjustments made in the manuscript, including the added paragraph on HIV-1 near full-length genome sequencing and the figure of HIV-1 reservoir in patients stratified by Fiebig stage at initiation of ART.

Reply: We highly appreciate the reviewers' positive assessment of the improvements in our manuscript.

References

- 1 Gossez M, Martin GE, Pace M, Ramjee G, Premraj A, Kaleebu P *et al.* Virological remission after antiretroviral therapy interruption in female African HIV seroconverters. *AIDS (London, England)* 2019; **33**: 185–197.
- 2 Muller V, von Wyl V, Yerly S, Boni J, Klimkait T, Burgisser P *et al.* African descent is associated with slower CD4 cell count decline in treatment-naive patients of the Swiss HIV Cohort Study. *AIDS (London, England)* 2009; **23**: 1269–1276.